

**1 Title page**

# Characterizing the nature and variability of avalanche hazard in western Canada

Bret Shandro, Pascal Haegeli

SFU Avalanche Research Program, Simon Fraser University, Burnaby, V5A 1S6, Canada

*Correspondence to*: Bret Shandro (bshandro@sfu.ca)

**2 Abstract**

The snow and avalanche climate types *maritime*, *continental* and *transitional* are well established and have been used extensively to characterize the general nature of avalanche hazard at a location, study interseasonal and large-scale spatial

variabilities and provide context for the design of avalanche safety operations. While researchers and practitioners have an experience-based understanding of the avalanche hazard associated with the three climate types, no studies have described the hazard character of an avalanche climate in detail. Since the 2009/10 winter, the consistent use of Statham et al.'s (2017) conceptual model of avalanche hazard in public avalanche bulletins in Canada created a new quantitative record of avalanche hazard that offers novel opportunities for addressing this knowledge gap. We identified typical daily avalanche hazard

situations using Self Organizing Maps (SOM) and then calculated seasonal prevalence values of these situations. This approach produces a concise characterization measure that is conducive to statistical analyses, but still provides a comprehensive picture that is informative for avalanche risk management due to its link to avalanche problem types. Hazard situation prevalence values for individual seasons, elevations bands and forecast regions provide unprecedented insight into the interseasonal and spatial variability of avalanche hazard in western Canada.

**3 Introduction**

Snow avalanches are a serious natural hazard in mountainous regions around the world, threatening communities, transportation corridors, critical infrastructure (e.g., hydroelectric dams, transmission and communication lines, pipelines) and resource extraction operations. In western countries, most people killed by avalanches are recreationists that voluntarily expose themselves to avalanche hazard while pursuing recreational activities in the uncontrolled backcountry, such as backcountry

skiing, mountain snowmobile riding and out-of-bounds skiing. During the last ten years, approximately 140 recreationists were killed in avalanches in North America and Europe every winter (Avalanche Canada, 2017; CAIC, 2017; Techel et al., 2016).





The physical risk from avalanches is managed with a combination of avalanche safety planning and operational avalanche safety programs (Canadian Avalanche Association, 2016; McClung and Schaerer, 2006). Avalanche safety planning identifies general exposure to long-term avalanche hazard and designs solutions to either eliminate the risk (e.g., avoidance through land use regulation) or reduce the risk to an acceptable level. Risk reduction may take the form of engineering solutions (e.g., snow

nets, deflection dams, avalanche sheds) and/or operational avalanche safety programs that manage acute avalanche risk in real-time through continuous monitoring of snow and weather conditions and choosing short-term mitigation measures (e.g., temporary closures, artificially triggering avalanches) to meet organizational objectives. Examples include avalanche safety operations whose aim is to keep a mountain road open, ski patrol programs securing ski runs at a resort, or mountain guides choosing safe backcountry terrain for guests. To provide public recreationalists who make their own avalanche safety decisions

with relevant information for winter backcountry travel, government and non-government avalanche safety agencies issue daily public avalanche forecasts.

The concept of snow and avalanche climates has been used extensively to describe the general snow and avalanche conditions in mountain ranges and offer context for the design of avalanche safety programs (Armstrong and Armstrong, 1986; LaChapelle, 1966; McClung and Schaerer, 2006; Mock and Birkeland, 2000; Mock et al., 2016). The existing literature

describes three main snow climate types. The *maritime* snow climate is characterized by relatively warmer temperatures, cloudy skies, continuous heavy snowfall resulting in a snowpack with few persistent weak layers. Avalanches mostly occur during or immediately following a storm and warmer temperatures promote rapid stabilization. Avalanche safety programs in maritime mountain ranges therefore primarily rely on weather observations to assess the likelihood and severity of storm snow avalanches. Examples of mountain ranges with a maritime snow climate include the Canadian Coast Mountains (Haegeli and

McClung, 2007), the Cascades and Sierra Nevada in the US (Mock and Birkeland, 2000), and the Japanese sea-side mountains (Ikeda et al., 2009). The *continental* snow climate exhibits colder temperatures, more frequent periods of clear skies and less snowfall, which produces a thinner snowpack that is conducive to the formation of depth hoar and persistent weak layers. In this snow climate, avalanches are more frequently associated with failures on persistent structural weaknesses within the snowpack (McClung and Schaerer, 2006). The systematic monitoring of snowpack weaknesses is therefore crucial for

effectively managing avalanche risk in a continental snow climate. Prominent examples of mountain ranges with a continental snow climate include North America's Rocky Mountains (e.g., Mock and Birkeland, 2000; Haegeli and McClung, 2007) and the Upper Himalaya (Sharma and Ganju, 2000). Mountain ranges that experience weather with both maritime and continental influences are described to have a *transitional* snow climate (also referred to as *intermountain* in the Unit
ed States). The transitional snow climate is characterized by large snowfalls and weaknesses in the snowpack that can persist

for weeks and months. These weaknesses are typically facet-crust combinations resulting from rain-on-snow events primarily early in the winter, or surface hoar layers that develop during extended periods of clear weather in the main winter months. Examples of mountain ranges with a transitional snow climate include Canada's Columbia Mountains (Haegeli and McClung, 2007), Utah's Wasatch Range, most parts of the European Alps (Rudolf-Miklau et al., 2015), and New Zealand's Craigeburn Range (McGregor, 1990).



While avalanche safety practitioners and researchers have a qualitative, experience-based understanding of the character of three snow and avalanche climate types, relatively little research has been conducted to explicitly quantify the nature of the avalanche hazard associated with the different snow and avalanche climate types. Building on previous research, Mock and Birkeland (2000) introduced a classification algorithm that objectively classifies the local snow and avalanche climate of

individual winter seasons based on weather data from high-elevation weather sites from December to March. The input parameters for the classification include mean air temperature, total rainfall, total snowfall, total snow water equivalent (SWE) and the derived average December snowpack temperature gradient. The authors derived classification thresholds by analyzing the variabilities of the input parameters at locations with established snow-climate classifications (Armstrong and Armstrong, 1986). The newly developed algorithm allowed Mock and Birkeland (2000) to examine spatial and interseasonal variabilities

in snow and avalanche climate characteristics in the western United States and explore the potential effects of El Nino Southern Oscillation and the Pacific Decadal Oscillation on avalanche conditions. However, their description of the effect was limited to stating that the phases of the atmospheric oscillations result in either a shift to more maritime or more continental snow climate characteristics without providing more detail about nature of that change. Haegeli and McClung (2007) subsequently used Mock and Birkeland's (2000) classification algorithm to examine variability in snow and avalanche hazard characteristics

in southwestern Canada. While their application of the approach provided useful general insights consistent with the results of Mock and Birkeland (2000), a parallel analysis of persistent snowpack weaknesses reported in the industrial avalanche safety information exchange of the Canadian Avalanche Association showed that it is possible to have substantially different snowpack structures and associated avalanche activity among winters that were assigned to the same snow and avalanche climate type. This result highlighted that the existing snow climate classifications has considerable limitations for informing

avalanche risk management practices. This is not surprising as seasonally summarized weather observations only have limited connections to the factors driving daily avalanche hazard. Instead, avalanche hazard is determined by particular sequences of weather events (Gruber et al., 2004). However, summarizing the weather events of a season in a way that is more informative of the resulting avalanche hazard conditions is challenging and has so far not been attempted.

In 2008, a group of Canadian and American avalanche forecasters and researchers developed a *conceptual model of avalanche*

*hazard* (CMAH) to describe the judgmental avalanche hazard assessment process (Statham et al., 2017). The CMAH identifies key components of avalanche hazard and structures them in a systematic workflow to provide a meaningful pathway for synthesizing available avalanche safety observations (weather, snowpack and avalanche observations), conceptualizing hazard conditions, and choosing appropriate risk treatment actions. A key component of the CMAH is the identification and characterization of *avalanche problems* (Haegeli et al., 2010; Lazar et al., 2012), which represent operational avalanche safety

concerns that emerge from the preceding weather and snowpack conditions. Avalanche hazard assessments typically include one or more avalanche problems, which are described in terms of their avalanche problem type, where they can be found in the terrain, the likelihood of associated avalanches, and the destructive size of these avalanches. The CMAH defines nine different avalanche problem types, which represent typical, repeatable patterns of avalanche hazard formation and evolution. Identifying the type of an avalanche problem is a critically important step in the hazard assessment process as it provides an



overarching filter that sets expectations and influences subsequent decisions about relevant types of observation and effective approaches for risk reduction. For example, wind slab avalanche problems are generally associated with wind events (with or without new snow) and typically produce small to medium sized avalanches that are limited to steeper slopes on the leeward side of ridgelines. The potential for wind slab avalanche can often be recognized by changes in the appearance of the snow

surface and surface snow hardness, and typical warning signs include hollow, drum-like sounds and/or shooting cracks. Wind slab avalanche problems usually stabilize within a few days and are best managed by recognizing and avoiding susceptible areas until they have stabilized (Haegeli et al., 2010). Persistent slab avalanche problems, on the other hand, are associated with structural weaknesses in the snowpack that formed earlier in the winter. Avalanche vary in size from medium to very large and may occur on gentle terrain. Often, no obvious surface clues exist, and snowpack tests are required to locate the

structural weakness in the snowpack. This type of avalanche problem tends to persist for weeks or longer and a lack of avalanche activity is not a reliable indicator of low hazard. Associated avalanches can be triggered remotely, and avalanche tend to release above the trigger. Persistent slab avalanche problems are best managed with very conservative terrain choices. Extra time should be given allow persistent avalanche problems to stabilize and new terrain should only be approached very cautiously (Haegeli et al., 2010). The CMAH including the concept of avalanche problems resonated well with Canadian

avalanche professionals and the approach was quickly adopted into operational hazard assessment practices, information systems and training programs (Statham et al., 2017).

Avalanche Canada integrated the CMAH into their production of public avalanche bulletins at the beginning of the winter of 2009/10 and since the winter 2011/12, all public avalanche bulletins in Canada have been structured according to the CMAH. The resulting dataset of avalanche problem assessments offers new opportunities for describing the nature of avalanche hazard

conditions of individual winters in more detail and expanding the concept of snow and avalanche climate. The direct link between avalanche problems and risk management strategies suggests that a seasonal characterization based on avalanche problems can provide a more insightful perspective than a classification based on meteorological factors. The objective of this study is a) to develop a meaningful summary description of avalanche hazard that facilitates quantitative analyses of spatial and seasonal variability, and b) to use this measure to expand our understanding of the temporal and spatial variability of

avalanche hazard in western Canada.

## 4 Methods

Western Canada is particularly suited for studying snow and avalanche climates as the three main mountain ranges exhibit a wide range of snow climate and related avalanche hazard characteristics (Fig. 1; Haegeli and McClung, 2007; McClung and Schaerer, 2006) . Our approach for deriving a meaningful description of seasonal avalanche hazard conditions that facilitates

quantitative analyses consisted of two steps. To reduce the dimensionality of the daily hazard assessments to a more manageable level, we first identified typical daily hazard situations that encapsulate the information of the daily hazard assessments in a more condensed fashion without losing valuable details. In the second step, we calculated seasonal prevalence



values of the typical hazard situations to provide a quantitative description of the seasonal hazard conditions. The resulting prevalence values then allow us to examine both spatial and seasonal variabilities in avalanche hazard conditions. To provide context and link our results to previous research, we also applied the Mock and Birkeland (2000) snow and avalanche climate classification algorithm to weather observations from a select number of high-elevation observation sites. Each of these steps

is described in more detail in the following sections.

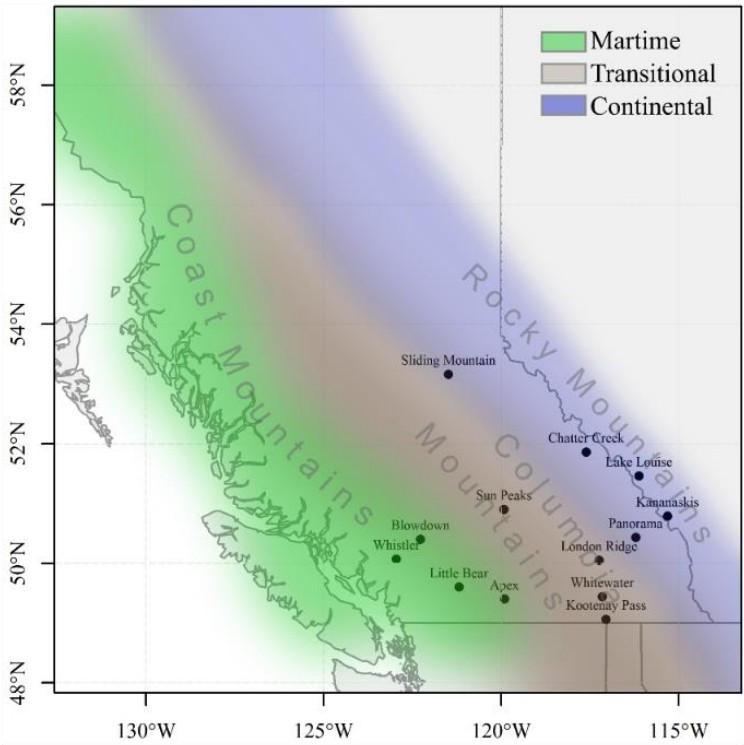

**Figure 1: General mountain ranges and snow climate zones of Western Canada with high elevation weather stations.**

## 4.1 Winter characterization based on avalanche problem assessments

### 4.1.1 Bulletin data

The primary dataset for this study consists of archived daily public avalanche bulletins from Avalanche Canada and Parks Canada, the two main agencies publishing public avalanche safety information in Canada from eight winter seasons (2009/10 to 2016/17). The core information presented in Canadian avalanche bulletins consists of a characterization of the existing avalanche problems according to the CMAH and predicted avalanched danger ratings for the three elevation bands alpine,

treeline and below treeline according to the five-point North American avalanche danger scale (Statham et al., 2010) for up to





three days into the future. Also included in the bulletin record (but hidden from bulletin users) are nowcast danger ratings that describe the severity of hazard in the three elevation bands at the time when the assessment was produced.

The combined dataset from Avalanche Canada and Parks Canada consists of 14,892 avalanche hazard assessments for 24 different forecast regions (Tables S1 & S2). Forecast regions that are only serviced with infrequent bulletins or bulletins of

reduced content (North Shore, North Rockies, Bighorn Country, Little Yoho, Whistler Blackcomb, and the Yukon forecast regions) were excluded to ensure a consistent analysis dataset. The final dataset for statistical analysis consisted of 13,396 public avalanche bulletin records spanning eight winters from 20 forecast regions.

Numerous adjustments to the boundaries of avalanche bulletin regions were made during the study period. During the winter seasons 2009/10 and 2010/11, Avalanche Canada produced public avalanche bulletins for six forecast regions (Fig. 2):

*Northwest*, *South Coast*, *North Columbia*, *South Columbia*, *Kootenay Boundary*, and *South Rockies*. In 2012, Avalanche Canada split some of their larger forecast regions to provide backcountry users with more targeted avalanche safety information (Fig. 2). The *South Coast* forecast region was separated into *Sea-to-Sky* and *South Coast Inland*, and the *Northwest* region was separated similarly into *Northwest Coastal* and *Northwest Inland*. In the Columbia Mountains, the *Cariboo* forecast region was split from the *North Columbia* forecast region and the *South Columbia* region was reduced to accommodate the new

*Purcell* forecast regions. In the Rocky Mountains, the *Lizard Range* was separated from the *South Rockies* forecast region. In 2015, Parks Canada separated the *Little Yoho* forecast region from the *Banff, Yoho, and Kootenay* region. The most recent change in the forecast regions occurred in 2017 when Avalanche Canada expanded the boundaries of the *North Shore* to include the mountains on the Sunshine Coast and along Howe Sound and renamed the region *South Coast*. The relatively recently created *Little Yoho* and *South Coast* forecast regions were not included in our analysis because of the brevity of their

use.




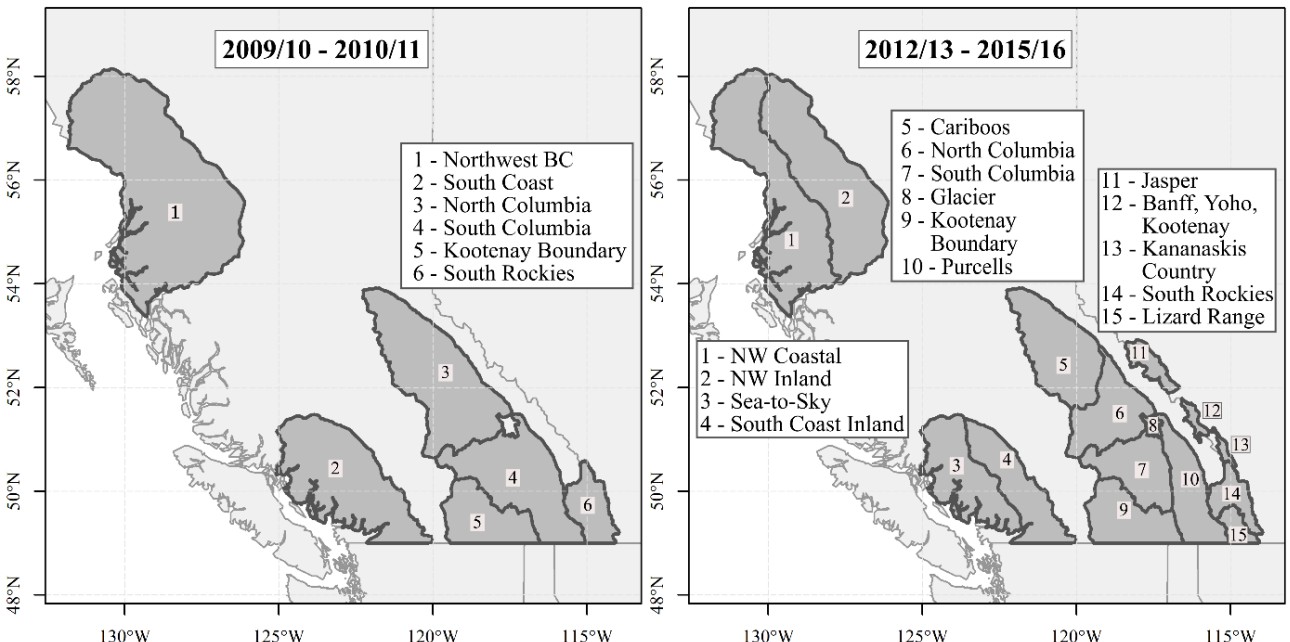

**Figure 2: Public avalanche bulletin regions for Western Canada from 2009/10 to 2015/16. Slight boundary modifications were made in the Sea-to-Sky region for the 2016/17 winter.**

For the present analysis, we focus on the main hazard dimensions described in the CMAH: the likelihood of avalanches and destructive size assessments of avalanche problems. In the CMAH, the likelihood of avalanches is expressed with an ordinal five-point scale that uses the qualitative terms *unlikely* (level 1), *possible* (2), *likely* (3), *very likely* (4) and *almost certain* (5). The destructive size of avalanches is described using the well-established destructive force classification system (CAA, 2014; AAA, 2016), which ranks avalanches on an ordinal five-point scale of exponential nature from *Size 1* (relatively harmless to

people) to *Size 5* (could destroy a village or a forest area of approx. 40 ha). However, since the Canadian bulletin writing software AvalX (Statham et al., 2012) allows forecasters to choose intermediate steps on these scales, both hazard dimensions are represented with ordinal nine-point scales in our dataset. For both likelihood of avalanches and destructive size, forecasters provide three estimates which include a typical value, which represents their best estimates for the likelihood of avalanches and destructive size under the existing conditions, as well as minimum and maximum values for both dimensions to express

uncertainty. To illustrate the overall hazard, the CMAH depicts the likelihood of avalanches and destructive size value triplets in a hazard chart (Fig. 3). Each avalanche problem is represented by a point plotted at the coordinates described by the typical value estimates, and a surrounding square that illustrates the associated uncertainty. These hazard charts have proven themselves to be effective for displaying and communicating the nature of avalanche hazard in a concise way among avalanche professionals. Additional information relevant for the present analysis include the specification of the elevation band to which

avalanche problem assessments relate to (*alpine*, *treeline* and *below treeline*) and the overall, elevation-band specific nowcast danger ratings (*no rating*, *low*, *moderate*, *considerable*, *high*, *extreme*).





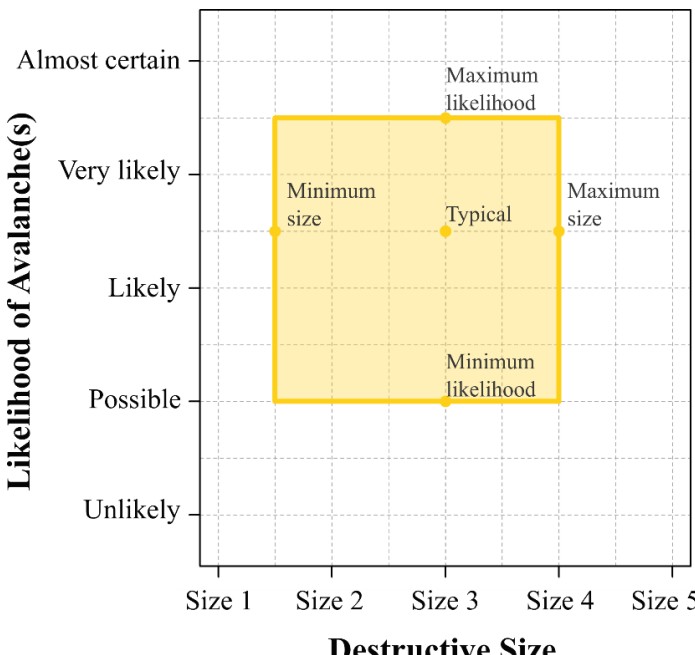

**Figure 3: Example hazard chart summarizing avalanche hazard conditions using the likelihood and destructive size assessments for an avalanche problem. The points in the middle are the typical value of likelihood and destructive size, while the outer edges are the maximum and minimum values which represent the associated uncertainty in estimations.**

### 4.1.2 Identifying typical hazard situations

Since the number of possible combinations of avalanche problems with different likelihoods of avalanches and destructive sizes is extremely large, we first used self-organizing maps (SOM; Kohonen 2001) to identify a smaller number of typical avalanche hazard situations and assign each hazard assessment to one of these situations. SOM is a type of unsupervised

artificial neural network widely applied to clustering problems (Kohonen, 2013). The method reduces multidimensional data by assigning the records of an input dataset to a prescribed number of $n$ nodes that are arranged in a two-dimensional map space. Each of these nodes is characterized by a weight vector $W$, which describes the nature of the node according to the parameters of the input dataset, and its position within the two-dimensional map space. At the beginning of the analysis, the map space is first initialized by randomly picking $n$ records from the input dataset and assigning each of their parameter vectors

to the weight vectors of one of the nodes. Subsequently, the parameter vector of each record of the input dataset (i.e., input vectors) is then placed onto the map by finding the node with the weight vector that resembles the input vector the most using a Euclidian distance measure. The node with the shortest Euclidean distance is termed the "*best matching unit*" (BMU). The map then "self organizes" by updating the nodes in the neighbourhood of the BMU by shifting their weight vectors closer to the input vector. This updating step is described by

$$W(t + 1) = W(t) + \Theta(t)\alpha(t)[V(t) - W(t)], \tag{1}$$



where *t* is the current iteration, *W* is the weight vector, *V* is the input vector, *Θ* is the neighbourhood function that considers distance from the BMU, and *α* is an iteration-dependent learning function. The SOM is trained by applying Equation 1 to each input vector in the dataset and the training limit specifies the number of iterations the training process is repeated. Following the training process, each SOM node reflects a typical pattern that emerged from the original dataset with neighbouring nodes

representing more similar patterns and nodes located further away in the map space featuring more distinct patterns.

Increasing map size results in more detailed patterns and more accurate results. However there is a trade-off between compressing information and accuracy of the SOM (Liu et al., 2006). To select a robust map size, we trained several SOMs while examining the relationship between quantization error and topographical error. Quantization error is a measure of internal node similarity and calculates average distance between each input vector for each node. Topographical error measures

the distance from best match node to second best matching node for each input vector. Readers interested in SOM are referred to Kohonen (2001) for a comprehensive description of the method.

The SOM analysis in this study was conducted with the Kohonen package in R (Wehrens and Buydens, 2007). The input data for the analysis were parameter vectors for each hazard assessment where each of the eight avalanche problem types was represented with a value triplet (minimum, typical, and maximum) for likelihood of avalanches and destructive size. If a

particular avalanche problem type existed in an assessment, its assessments on the ordinal likelihood of avalanches and destructive size scales were represented by numerical values between 1 and 9. If an avalanche problem type did not exist, the values for its three likelihood and destructive size variables were all set to zero. In the rare occasion that an avalanche problem type existed more than once in the same assessment (e.g., multiple persistent slab avalanche problems), only the more severe problem was included in the analysis. This resulted in a training dataset for the SOM analysis of 38,982 assessments from all

elevation bands with 51 variables (8 x 6 parameters to characterize the hazard conditions, elevation band information, assessment ID and bulletin region ID).

To facilitate the interpretation of the identified typical hazard situations (i.e., SOM nodes), we calculated the frequency of the avalanche problem types, the median hazard chart and the distribution of the nowcast avalanche danger ratings from the hazard assessments assigned to the particular node. The median hazard chart visualizes the median likelihood of avalanches and

destructive size value triplets (minimum, typical, maximum) for avalanche problems occurring in more than 50% of the associated assessments. To further examine differences in the severity of typical hazard situations, we used pairwise Wilcoxon rank-sum tests to evaluate differences between danger rating distributions of typical hazard situations. Pearson's chi-squared tests were used to examine differences in the prevalence in typical hazard situations among elevation bands.

### 4.1.3 Typical hazard situation prevalence

To provide seasonal avalanche hazard characterizations that offer a more comprehensive perspective and are suitable for quantitative analyses, we calculated the prevalence of each typical hazard situation for each winter season (December 1 to April 15) for the entire dataset. The resulting seasonal characterization consists of a set of twelve avalanche hazard situation prevalence percentage values that add up to 100%. The time period from December 1 to April 15 was chosen because avalanche



bulletins are consistently published for all forecast regions during that period. To better highlight variations in avalanche hazard conditions, we calculated seasonal anomaly values for the hazard situation prevalence. Due to the missing of Park Canada bulletins for the first two winter seasons (2009/10 and 2010/11), we calculated the annual prevalence anomalies in two different ways:

- Only using bulletin information from Avalanche Canada forecast regions to calculate overall means and seasonal anomalies for the entire study period.
- Using bulletin information from both Avalanche Canada and Parks Canada to calculate overall means and seasonal anomalies for the 2011/12 to 2016/17 winter seasons.

While the first perspective provides insight into variations for the entire study period, it is limited to the bulletin regions of
Avalanche Canada (i.e., Coast Mountains, Columbia Mountains, Southern Rocky Mountains). The second perspective offers a more comprehensive perspective as it also includes the Parks Canada forecast regions (primarily located in the central Rocky Mountains), but it is only available for the last six winter seasons. To examine spatial variability in avalanche hazard conditions, we calculated avalanche hazard situation prevalence values for individual elevation bands and forecast regions and compared them to the average prevalence values of the complete dataset.

**4.2 Winter characterization based on weather observations**

To create a baseline characterization of avalanche winters and tie our results back to the existing literature on snow and avalanche climates, we applied the snow climate classification scheme of Mock and Birkeland (2000) to a select number of high-elevation weather sites in western Canada. High elevation automated weather sites with consistent daily weather and snowpack observations (including height of snowpack, 24 hr new snow, rain) from early December to late March are rare in
western Canada. Available weather records from Environment Canada, Parks Canada, the avalanche program of the British Columbia Ministry of Transportation and Infrastructure and the InfoEx (industrial information exchange among avalanche safety programs in Canada administered by the Canadian Avalanche Association) were scanned for suitable weather sites. Our search revealed 13 suitable weather stations (Fig. 1 & Table S3), for which we retrieved daily records of mean air temperature (°C), total rainfall (mm per 24 hours), total snowfall (cm per 24 hours), total snow water equivalent (SWE, mm per 24 hours)
and height of snowpack (cm).

Since the available meteorological data did not have all the parameters required for the Mock and Birkeland (2000) classification, some of them had to be derived. Our method for this closely followed the approach taken by Haegeli and McClung (2007). The SWE values for Environment Canada stations were estimated from daily snowfall records by assuming a seasonal average new snow density of 100 kg/m$^3$. For the Ministry of Transportation and Infrastructure (MOTI) data, we
calculated daily summaries from 6-hourly observations. The daily rainfall was approximated by subtracting the SWE of new snow from values of total precipitation (Hägeli and McClung, 2003). To calculate the December temperature gradient, we assumed basal snowpack temperature of 0°C and divided the mean December air temperature by the average December snow



depth (Mock and Birkeland, 2000). Records from stations that were missing a variable continuously for more than 10 days were not used for the seasonal snow climate classification.

## 5 Results

### 5.1 Typical hazard situations

5  The topographical and quantization error for various SOM grid sizes (Fig. 4) showed that the topographic error is constant and independent of grid size while a considerable marginal reduction in the quantization error can be seen with increasing grid size. Balancing cluster error and interpretability of the emerging clusters, we selected a 4×3 grid (i.e., 12 nodes) with hexagonal arrays and a training length of 200 iterations for the final SOM analysis. Our analysis, therefore, identified twelve typical avalanche hazard situations, and each assessment in our dataset was assigned to one of these situations (Table 1). Hazard

10  assessments that contained no avalanche problems were automatically assigned into an additional *No avalanche problems hazard situation* class separate from the SOM analysis (Fig. 5a).

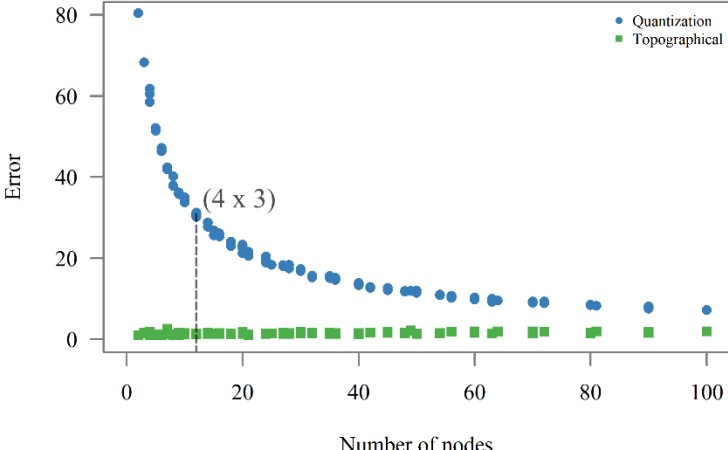

**Figure 4: Quantization and topographical errors for SOM at various grid sizes.**

**Table 1: Elevation band specific distribution of SOM classified typical hazard situations.**

| Hazard Situation | Overall | | No danger ratings | | Danger ratings[A] | | | Alpine | | Treeline | | Below treeline | |
|---|---|---|---|---|---|---|---|---|---|---|---|---|---|
| | N | (%) | N | (%) | 1Q | Median | 3Q | N | (%) | N | (%) | N | (%) |
| *No avalanche problems* | 5862 | (15) | 232 | (4) | 1 | 1 | 1 | 122 | (1) | 667 | (5) | 5073 | (38) |
| *Loose dry avalanche* | 1447 | (4) | 4 | (<1) | 1 | 2 | 2 | 493 | (4) | 520 | (4) | 434 | (3) |
| *Wind slab* | 4492 | (11) | 105 | (2) | 2 | 2 | 2 | 2517 | (19) | 1771 | (13) | 204 | (2) |
| *Storm slab* | 4475 | (11) | 79 | (2) | 2 | 3 | 3 | 1267 | (10) | 1957 | (15) | 1251 | (9) |
| *Storm & wind slab* | 1448 | (4) | 5 | (<1) | 3 | 3 | 3 | 765 | (6) | 631 | (5) | 52 | (<1) |
| *Storm & persistent slab* | 3643 | (9) | 8 | (<1) | 2 | 3 | 3 | 1338 | (10) | 1419 | (11) | 886 | (7) |
| *Storm & deep persistent slab* | 1483 | (4) | 0 | (0) | 3 | 3 | 3 | 674 | (5) | 632 | (5) | 177 | (1) |
| *Storm, wind, & persistent slab* | 1058 | (3) | 3 | (<1) | 3 | 3 | 4 | 455 | (3) | 586 | (4) | 17 | (<1) |
| *Persistent slab* | 3141 | (8) | 9 | (<1) | 2 | 2 | 3 | 512 | (4) | 706 | (5) | 1923 | (15) |
| *Persistent slab plus* | 4766 | (12) | 14 | (<1) | 2 | 3 | 3 | 2395 | (18) | 2258 | (17) | 113 | (1) |
| *Deep persistent slab* | 3572 | (9) | 57 | (2) | 2 | 2 | 3 | 1665 | (13) | 1425 | (11) | 482 | (4) |
| *Spring-like* | 3068 | (8) | 258 | (8) | 1 | 2 | 2 | 641 | (5) | 813 | (6) | 1614 | (12) |
| *Loose wet & persistent slab* | 1085 | (3) | 4 | (<1) | 2 | 2 | 3 | 336 | (3) | 501 | (4) | 248 | (2) |
| Overall | 39540 | | 778 | | | | | 13180 | | 13180 | | 13180 | |

A    Key for numerical danger ratings scale: Low (1), Moderate (2), Considerable (3), High (4), Extreme (5)



Two typical hazard situations emerged from the SOM analysis that generally represent low hazard conditions during the main winter months. The *Loose dry avalanche* hazard situation (Fig. 5b) included mostly of dry loose avalanche problems but had a substantial contribution from wind slab avalanche problems. Overall, this hazard situation had the lowest danger ratings among all hazard situation types. The pure *Wind slab* hazard situation (Fig. 5c) consisted of assessments with relatively low

likelihood of small avalanches and therefore also exhibited low danger ratings. These two low hazard situations made up 4% and 11% of our complete dataset. While the *Loose dry avalanche* hazard situation was assigned evenly in all elevation bands, the pure *Wind slab* hazard situation was much more dominant in the alpine and at treeline (19% and 13%) than below treeline (2%).

Five distinct hazard situations were identified for hazard assessments that predominantly contain a storm slab avalanche

problem. The pure *Storm slab* hazard situation (Fig. 5d) was the classification for assessments with only a storm slab avalanche problem. This is reflected in the lowest danger ratings of the five storm slab avalanche problem situations (Wilcoxon rank-sum test: p-value < 0.001). This hazard situation occurred more frequently at treeline (15%) than in the alpine and at below treeline (10% and 9% respectively). The addition of a wind slab avalanche problem made the *Storm & wind slab* hazard situation (Fig. 5e) significantly more severe. Consistent with the pure *Wind slab* hazard situation, the *Storm & wind slab* hazard

situation was observed more frequently in the alpine and at treeline. The *Storm & deep persistent slab* hazard situation (Fig. 5g) was more severe than the *Storm & persistent slab* (Fig. 5f), but the *Storm, wind & persistent slab* hazard situation (Fig. 5h) was the most severe of all the storm slab hazard situations. All of these three hazard situations occurred approximately 5% in the alpine and at treeline, but they were rarely observed below treeline.

Three of the identified hazard situations were dominated by persistent weaknesses in the snowpack. The *Persistent slab* and

*Persistent slab plus* hazard situations (Fig. 5i & j) were both characterized by persistent slab avalanche problems, but they differed in their severity. Despite having similar median hazard charts, the danger ratings of the *Persistent slab plus* situation were significantly higher (median value 3 versus 2; Wilcoxon rank-sum test: p-value < 0.001). Deep persistent and wind slab avalanches problems were most common for the *Deep persistent slab* hazard situation (Fig. 5k). While the *Persistent slab* hazard situation was most prominent below treeline (15%), the other two situations were more frequently assigned in the alpine

and at treeline. It is noteworthy that all hazard situations with persistent weaknesses frequently included wind slab avalanche problems.

The last two typical hazard situations identified by the SOM analysis represent conditions that generally occur during warmer temperatures. The *Spring-like* hazard situation (Fig. 5l) primarily consisted of wet loose avalanches and wet slab avalanches. As expected, this hazard situation was significantly more prevalent below treeline (12%) than above (5% in alpine and 6% at

treeline) (Chi-square test: p-value < 0.001). This hazard situation also had the highest percentage of assessments that did not have a danger rating associated with it (8%). The *Loose wet & persistent slab* hazard situation (Fig. 5m) typically occurred during periods of warm wet weather caused by atmospheric river events, which can occur anytime during a winter (Spry et al., 2014).





**Figure 5a-e: Hazard characteristics of the 12 typical hazard situations including the avalanche problem distribution, median hazard char, and danger rating distribution.**





**Figure 5f-j: Hazard characteristics of the 12 typical hazard situations including the avalanche problem distribution, median hazard char, and danger rating distribution.**





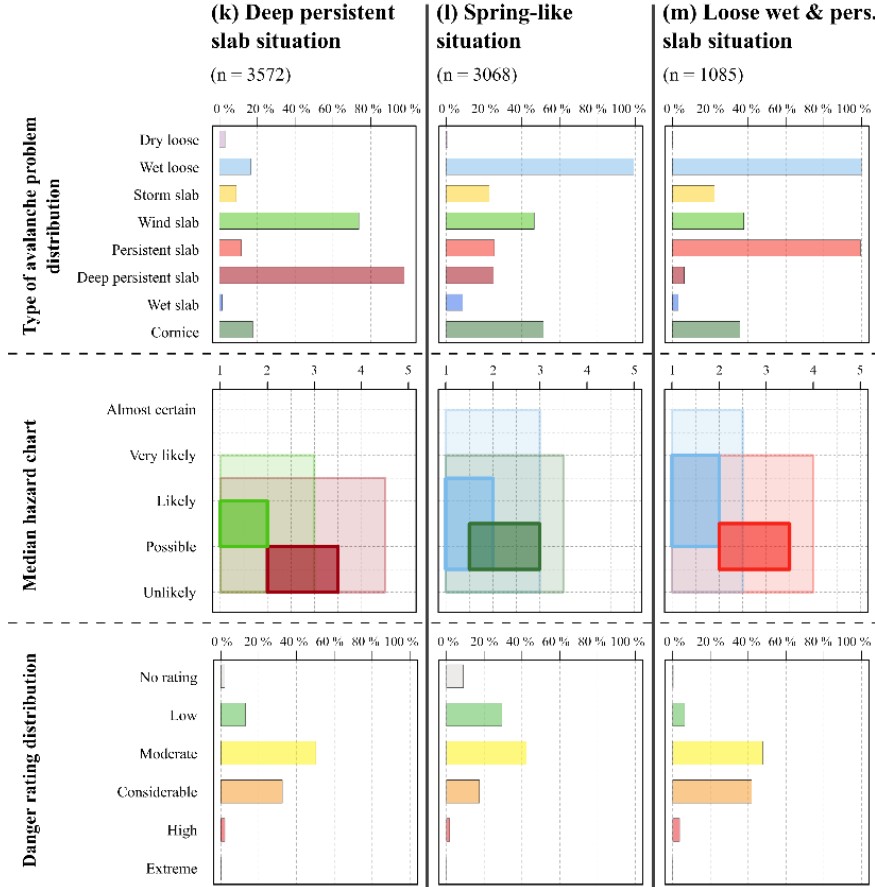

**Figure 5k-m: Hazard characteristics of the 12 typical hazard situations including the avalanche problem distribution, median hazard char, and danger rating distribution.**

Over the entire study period, including all forecast regions and all elevation bands, the most common hazard situations were the *Persistent slab plus* hazard situation (12% of assessments), the pure *Wind slab* hazard situation (11%) and the pure *Storm slab* hazard situations (11%). These hazard situations were closely followed by the *Storm & persistent slab* hazard situation (9%), *Deep persistent slab* hazard situation (9%), *Persistent slab* hazard situation (8%), and *Spring-like* hazard situation (8%). The prevalence of all other hazard situations was less than 5%.

A comparison of hazard situation prevalence values among the three elevation bands revealed that the avalanche conditions below treeline were dominated by the *No avalanche problems* hazard situations, which account for 38% assessments. When there was an avalanche problem, the conditions were relatively simple and of lower severity. The three avalanche hazard situations *Storm slab* hazard situation (9%), *Persistent slab* hazard situation (15%) and *Spring-like* hazard situation (12%) accounted for an additional 36% of the assessments. The distribution of the typical hazard situations in the alpine and treeline

elevation bands exhibited considerable similarity. The most common hazard situations in these elevation band were pure *Wind*





*slabs* (19% in alpine and 13% at treeline) and the more severe *Persistent slab plus* hazard situation (18% and 17%). The three hazard situations *Storm slabs*, *Storm & persistent slabs* and *Deep persistent slabs* combined were responsible for another 30% of the hazard situations in the alpine and at treeline. These results nicely illustrate that avalanche hazard situations in the alpine and at treeline are more complex and more varied than below treeline.

## 5.2 Winter characterization

### 5.2.1 Typical hazard situation prevalence: Seasonal variability

The analysis of the seasonal hazard situation prevalence revealed substantial winter-to-winter variabilities (Fig. 6). During the winter seasons when bulletins were available from both Avalanche Canada and Parks Canada (2011/12 to 2016/17; Fig. 7), the 2012/13 and 2014/15 winters were most normal (i.e., most similar to overall averages). The winter of 2011/12 was characterized by a higher prevalence of *Storm & wind slab* and *Storm, wind & persistent slab* hazard situations (+8 and +7 percentage points relative to 2011/12–2016/17 average) at the expense of the equivalent hazard situations without wind slab avalanche problems (i.e., *Storm slab* and *Storm & persistent slab* hazard situations). The 2013/14 winter was dominated by the presence of a deep persistent avalanche problem, which resulted in increased prevalence of *Deep persistent* and *Storm & deep persistent* hazard situations (+5 and +8 percentage points) and fewer *Wind slab* hazard situations (-6 percentage points). The winter of 2015/16 saw an additional 6 percentage points of *Storm slab* hazard situations, while the prevalence of *Deep persistent slab* hazard situations was 6 percentage points lower. The 2016/17 winter was substantially different again as it was characterized by more *Wind slab* hazard situations (+5 percentage points), more *Deep persistent slab* situations (+5 percentage points) and fewer *Spring-like* hazard situations (-4 percentage points).

Among the two winters when bulletins were only available from Avalanche Canada (Fig. 8), the 2009/10 winter stands out due to its extremely high prevalence of *Persistent slab* avalanche hazard situations (+21 percentage points relative to overall average with Avalanche Canada bulletins only). The winter of 2010/11 exhibited an increase in *Storm & wind slab* and *Storm, wind & persistent slab* hazard situations similar to the 2011/12 winter (+10 and +8 percentage points), but this time it was due to a lower prevalence of *Storm slab* and *Storm & persistent slab* hazard situations. While the lack of Parks Canada bulletins could at least partially be responsible for the lower prevalence of persistent slab related hazard situations during the 2011/12 winter, it cannot explain the extremely high prevalence of *Persistent slab* avalanche hazard situations in the 2009/10 winter. The similarities in the anomaly patterns for the winters 2011/12 to 2016/17 with and without the Parks Canada bulletins further support the conclusion that the observed patterns for the first two winters in the study period are meaningful representations of the overall hazard conditions.





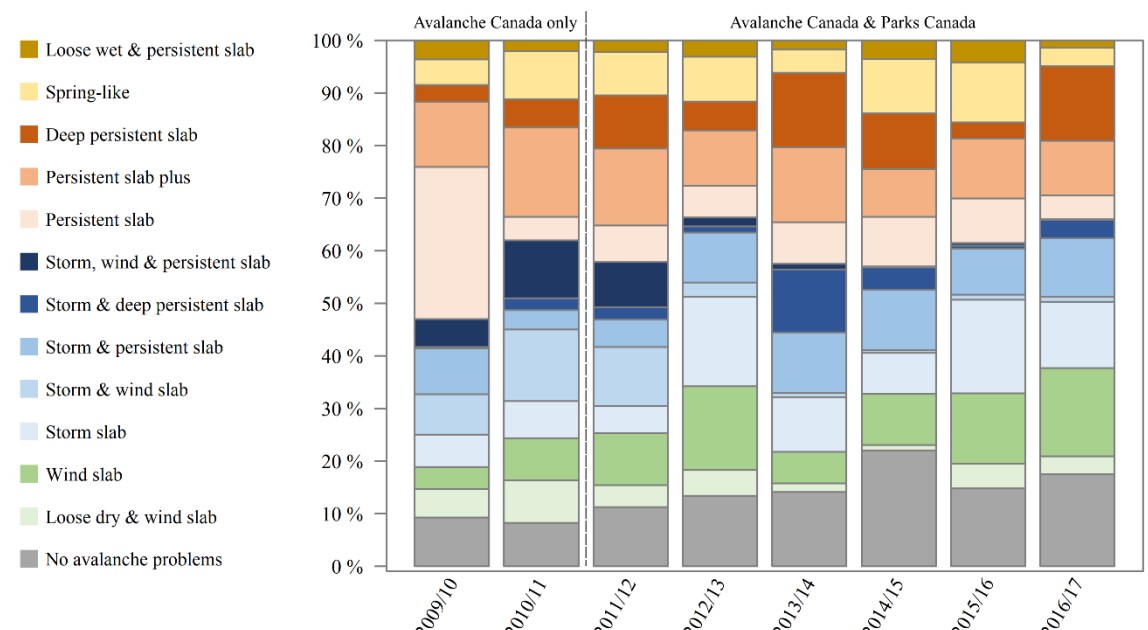

**Figure 6: Seasonal prevalence of typical hazard situations.**

| | | | | | Yearly anomaly | | | | |
|---|---|---|---|---|---|---|---|---|---|
| **Hazard situation** | **Mean** | **2009 /10** | **2010 /11** | **2011 /12** | **2012 /13** | **2013 /14** | **2014 /15** | **2015 /16** | **2016 /17** |
| *No avalanche problems* | 15 | n/a | n/a | -4 | -2 | -1 | 7 | -1 | 2 |
| *Loose dry avalanche* | 3 | n/a | n/a | 1 | 2 | -2 | -2 | 1 | 0 |
| *Wind slab* | 12 | n/a | n/a | -2 | 4 | -6 | -2 | 1 | 5 |
| *Storm slab* | 12 | n/a | n/a | -7 | 5 | -1 | -4 | 6 | 1 |
| *Storm & wind slab* | 3 | n/a | n/a | 8 | 0 | -2 | -2 | -2 | -2 |
| *Storm & persistent slab* | 10 | n/a | n/a | -4 | 0 | 2 | 2 | -1 | 2 |
| *Storm & deep persistent slab* | 4 | n/a | n/a | -2 | -3 | 8 | 0 | -3 | 0 |
| *Storm, wind, & persistent slab* | 2 | n/a | n/a | 7 | 0 | -1 | -2 | -1 | -2 |
| *Persistent slab* | 7 | n/a | n/a | 0 | -1 | 1 | 2 | 1 | -3 |
| *Persistent slab plus* | 12 | n/a | n/a | 3 | -1 | 3 | -3 | 0 | -1 |
| *Deep persistentslab* | 10 | n/a | n/a | 1 | -4 | 5 | 1 | -6 | 5 |
| *Spring-like* | 8 | n/a | n/a | 0 | 1 | -3 | 3 | 4 | -4 |
| *Loose wet & persistent slab* | 3 | n/a | n/a | -1 | 0 | -1 | 1 | 1 | -1 |

**Figure 7: Typical hazard situation prevalence in percent for all of Western Canada from 2011/12 to 2016/17, overall mean and winter season anomalies. Red shading denotes positive anomalies, and blue shading indicates negative anomalies greater than 5 percentage points.**





| Hazard situation | Mean | Yearly anomaly | | | | | | | |
|---|---|---|---|---|---|---|---|---|---|
| | | 2009 /10 | 2010 /11 | 2011 /12 | 2012 /13 | 2013 /14 | 2014 /15 | 2015 /16 | 2016 /17 |
| *No avalanche problems* | 15 | -5 | -6 | -4 | -2 | -1 | 8 | 1 | 3 |
| *Loose dry avalanche* | 3 | 3 | 5 | 1 | 1 | -2 | -2 | 0 | -1 |
| *Wind slab* | 12 | -8 | -4 | -1 | 4 | -6 | -1 | 3 | 6 |
| *Storm slab* | 12 | -6 | -5 | -7 | 6 | -1 | -4 | 8 | 2 |
| *Storm & wind slab* | 4 | 4 | 10 | 9 | -1 | -4 | -3 | -3 | -3 |
| *Storm & persistent slab* | 10 | -1 | -6 | -5 | 1 | 3 | 2 | 0 | 3 |
| *Storm & deep persistent slab* | 4 | -4 | -2 | -2 | -3 | 10 | 0 | -3 | 0 |
| *Storm, wind, & persistent slab* | 3 | 2 | 8 | 7 | -1 | -3 | -3 | -2 | -3 |
| *Persistent slab* | 8 | 21 | -3 | -1 | -2 | 0 | 1 | -1 | -3 |
| *Persistent slab plus* | 13 | 0 | 4 | 4 | -2 | 2 | -2 | -2 | -1 |
| *Deep persistent slab* | 7 | -3 | -1 | 0 | -2 | 6 | 0 | -4 | 4 |
| *Spring-like* | 8 | -3 | 1 | 1 | 1 | -3 | 3 | 3 | -5 |
| *Loose wet & persistent slab* | 3 | 1 | -1 | -1 | 1 | -1 | 1 | 2 | -1 |

**Figure 8: Typical hazard situation prevalence in percent for Avalanche Canada forecast regions over entire study period, overall mean and winter season anomalies. Red shading denotes positive anomalies, and blue shading indicates negative anomalies great than 5 percentage points.**

### 5.2.2 Typical hazard situation prevalence: Spatial variability

The prevalence of typical hazard situations in the 15 forecast regions provides insight into the regional differences in the avalanche hazard conditions over the last six winters of our study period. The regional mean prevalence values for the three elevations bands are shown graphically in **Error! Reference source not found.**, whereas the associated regional anomalies

are presented in Fig. 10.

In the alpine (Fig. 9-10), the *Sea-to-sky,* and the *South Coast* were characterized by a substantially higher prevalence of pure *Wind slab* and *Strom slab* hazard situations, which was at the expense of *Persistent slab plus* and *Deep persistent slab* hazard situations. Somewhat surprisingly, the *Northwest Coast* also showed a positive anomaly for the *Storm & persistent slab* hazard situation, which was compensated by negative anomalies in all other hazard situations involving persistent weaknesses, while

the *Northwest Inland* exhibited noticeably higher prevalence of *Deep persistent slab* situations in exchange for less *Storm & persistent slab* situations. The interior regions generally showed negative anomalies for the pure *Wind slab* and pure *Storm slab* hazard situations. In the *Cariboo* and the *North and South Columbia* regions, this was compensated with a higher prevalence of *Storm & persistent slab* hazard situations. The *Cariboo* also exhibited a higher prevalence of *Persistent slab plus* hazard situations. In *Glacier*, the decrease in pure Wind slab hazard situations was compensation by an increase in *Storm*

*& wind slab* situations. *Glacier* also showed a higher prevalence of *Loose dry avalanche* hazard situations and a lower prevalence of *deep persistent slab* situations. *Kootenay Boundary* was the forecast region with the hazard situation prevalence values most closely to the overall means, the only exception being having markedly lower *Deep persistent slab* prevalence values. The main feature of the *Purcells* and the *South Rockies* was the higher prevalence of pure *Persistent slab plus* hazard situations at the cost of fewer pure *Storm slabs*, *Storm & persistent slab* and *Wind slab* hazard situations (*Purcells* only). *Banff,*

*Yoho, and Kootenay (BYK), Jasper,* and *Kananaskis Country* exhibited a similar pattern, but higher prevalence was completely



dominated by *Deep persistent slab* hazard situations (+19 to 28 percentage points). The *Lizard Range* only experienced fewer *Deep persistent slab* situations and a slightly lower prevalence of *Storm & persistent slab* hazard situations and a slightly higher prevalence of pure *Wind slabs*.

The patterns observed at treeline generally mirrored the observations described for the alpine (Fig. 9-10), but the magnitude of the patterns varied slightly. Three forecast regions in the southeastern part of the study area were generally close to overall means. The only exceptions were fewer *Deep persistent slab* situations in the *Kootenay Boundary* and *Lizard Range* and greater prevalence of *Persistent slab plus* situations in the *Purcells*.

The regional patterns we found below treeline varied considerably from the alpine and treeline (Fig. 9-10). The *Northwest Coast* was characterized by a higher prevalence of *Storm & wind slab* and *Storm & persistent slab* situations, which was compensated by fewer assessments with *No avalanche problems*. The *Northwest Inland*, and *Lizard Range*, were the forecast regions with the hazard situation prevalence values most closely to the overall means. The *Sea-to-Sky* and *South Coast - Inland* area exhibited a higher prevalence of pure *Storm slab* situations and *No avalanche problem* situations which was offset with fewer *Persistent slab* situations. All interior hazard areas had considerably fewer assessments with *No avalanche problems*. In the *Cariboo, North Columbia,* and *South Columbia* regions, *Persistent slab* situations and *Storm & persistent slab* situations were responsible for approximately one-third of all assessments. *Glacier* had a higher prevalence of pure *Storm slab* situations, and in turn fewer situations with persistent slab problems. Aside from having fewer *No avalanche problem* situations, the hazard situation prevalence values for *Kootenay Boundary* were found to be close to overall mean values. The *BYK, Jasper, and Kananaskis Country* forecast regions had more situations with *No avalanche problems*, which was compensated with fewer pure *Storm slab* and *Storm & persistent slab* hazard situations. *Jasper* and *Kananaskis Country* were characterized with negative anomalies for *Persistent slab* situations, while both *Jasper* and *BYK* regions showed higher prevalence values for *Deep persistent slab* situations. It is worth highlighting that *No avalanche problems* hazard situations were much more prevalent in the Rocky Mountain regions than the Columbia Mountains and most of the Coast Mountain forecast regions.



**Figure 9: Forecast region hazard situation prevalence in forecast regions in the alpine, treeline, and below treeline elevation bands from 2011/12 to 2016/17.**



| HazardSituation | Mean | Northwest Coastal | Northwest Inland | Sea-to-Sky | South Coast -Inland | Cariboos | Monashees | South Columbia | Glacier | Kootenay Boundary | Purcells | Banff, Yoho and Kootenay | Jasper | Kananaskis Country | South Rockies | Lizard Range |
|---|---|---|---|---|---|---|---|---|---|---|---|---|---|---|---|---|
| **Alpine** | | | | | | | | | | | | | | | | |
| *No avalanche problems* | 5 | -3 | -1 | 5 | 3 | -1 | 1 | 1 | -3 | -1 | -1 | 2 | -4 | -3 | 2 | 2 |
| *Loose dry avalanche* | 3 | -2 | -3 | -1 | -2 | -1 | 0 | 0 | 6 | -2 | -2 | 4 | 6 | 1 | -1 | -1 |
| *Wind slab* | 14 | 1 | 8 | 7 | 6 | -3 | -2 | -2 | -5 | 3 | -2 | -6 | -3 | -7 | 3 | 2 |
| *Storm slab* | 10 | 6 | -4 | 12 | 8 | -2 | -1 | 1 | -4 | 2 | -5 | -7 | -6 | -6 | -1 | 5 |
| *Storm & wind slab* | 4 | 3 | 0 | 4 | 1 | -1 | -2 | -1 | 4 | 0 | -2 | -3 | -2 | -2 | -1 | 1 |
| *Storm & persistent slab* | 11 | 7 | -6 | -4 | -2 | 9 | 11 | 10 | 5 | 5 | 4 | -9 | -9 | -6 | -9 | -5 |
| *Storm & deep persistent slab* | 5 | 2 | 4 | -1 | -2 | 1 | -1 | -2 | -1 | -1 | 0 | -2 | -1 | -1 | 2 | 2 |
| *Storm, wind & persistent slab* | 3 | -2 | -2 | -1 | 0 | 1 | 3 | 1 | 4 | 0 | 1 | -2 | -2 | -1 | -1 | -1 |
| *Persistent slab* | 5 | -4 | -3 | -2 | -2 | -2 | -1 | -1 | 6 | -2 | 4 | 6 | 0 | 8 | -3 | -3 |
| *Persistent slab plus* | 17 | 2 | 2 | -13 | -5 | 6 | 1 | 1 | -5 | -2 | 7 | -7 | 2 | 3 | 8 | 1 |
| *Deep persistent slab* | 12 | -7 | 7 | -9 | -8 | -7 | -8 | -8 | -8 | -7 | -3 | 25 | 23 | 16 | 0 | -6 |
| *Spring-like* | 6 | -1 | 0 | 4 | 2 | -2 | -2 | -1 | -2 | 3 | -2 | 1 | -3 | -2 | 3 | 4 |
| *Loose wet & persistent slab* | 4 | -1 | -3 | 0 | 0 | 1 | 1 | 1 | 2 | 2 | 2 | -2 | -1 | 0 | -2 | -1 |
| **Treeline** | | | | | | | | | | | | | | | | |
| *No avalanche problems* | 1 | -1 | 1 | 1 | 1 | 0 | 0 | 1 | -1 | 0 | -1 | 1 | -1 | -1 | 0 | 0 |
| *Loose dry avalanche* | 4 | -2 | -4 | -2 | -2 | -1 | 0 | -1 | 6 | -2 | -2 | 5 | 6 | 4 | -2 | -2 |
| *Wind slab* | 20 | 1 | 7 | 11 | 10 | -4 | -1 | 0 | -7 | 3 | -4 | -7 | -6 | -13 | 4 | 5 |
| *Storm slab* | 11 | 8 | -5 | 12 | 7 | -1 | 0 | 1 | -3 | 2 | -6 | -8 | -6 | -7 | -2 | 5 |
| *Storm & wind slab* | 4 | 2 | 1 | 4 | 1 | -2 | -2 | -2 | 7 | -1 | -2 | -3 | -2 | -1 | -2 | 0 |
| *Storm & persistent slab* | 11 | 6 | -6 | -3 | -2 | 8 | 10 | 9 | 5 | 5 | 4 | -9 | -9 | -4 | -8 | -5 |
| *Storm & deep persistent slab* | 6 | 1 | 5 | -1 | -1 | 1 | -1 | -1 | -2 | -1 | -1 | -2 | -1 | -1 | 2 | 2 |
| *Storm, wind & persistent slab* | 3 | -1 | -2 | -2 | -1 | 2 | 2 | 2 | 3 | 0 | 2 | -2 | -2 | -1 | -1 | 0 |
| *Persistent slab* | 3 | -3 | -2 | -1 | -1 | -1 | 0 | 0 | 5 | 0 | 2 | 1 | 2 | 3 | -2 | -1 |
| *Persistent slab plus* | 17 | -2 | -2 | -12 | -7 | 6 | 0 | 1 | -3 | 0 | 9 | -2 | -1 | 4 | 8 | 0 |
| *Deep persistent slab* | 13 | -8 | 9 | -10 | -9 | -7 | -9 | -10 | -10 | -9 | -2 | 28 | 25 | 19 | 1 | -7 |
| *Spring-like* | 5 | 0 | 0 | 4 | 2 | -1 | -2 | -1 | -2 | 1 | -2 | 0 | -3 | -3 | 3 | 3 |
| *Loose wet & persistent slab* | 3 | 0 | -2 | 0 | 1 | 1 | 1 | 1 | 1 | 2 | 1 | -1 | -2 | 0 | -1 | -1 |
| **Below treeline** | | | | | | | | | | | | | | | | |
| *No avalanche problems* | 40 | -10 | 3 | 8 | 7 | -10 | -12 | -12 | -12 | -9 | -8 | 14 | 10 | 24 | 6 | 0 |
| *Loose dry avalanche* | 3 | -2 | -3 | -1 | -2 | -1 | 0 | -1 | 5 | -1 | -2 | 1 | 7 | 2 | -2 | -1 |
| *Wind slab* | 1 | 3 | 4 | -1 | 0 | 0 | 0 | 0 | -1 | 0 | -1 | -1 | 0 | 0 | -1 | 0 |
| *Storm slab* | 15 | 12 | -2 | 12 | 6 | 0 | 0 | 2 | 5 | 2 | -4 | -12 | -10 | -13 | -2 | 1 |
| *Storm & wind slab* | 0 | 0 | 1 | 0 | 0 | 0 | 0 | 0 | 0 | 0 | 0 | 0 | 0 | 0 | 0 | 0 |
| *Storm & persistent slab* | 7 | 1 | -4 | -3 | -1 | 6 | 7 | 6 | 4 | 3 | 4 | -6 | -6 | -7 | -4 | -1 |
| *Storm & deep persistent slab* | 1 | -1 | 1 | -1 | -1 | 1 | 2 | 1 | -1 | 1 | 0 | -1 | -1 | -1 | 1 | 1 |
| *Storm, wind & persistent slab* | 0 | 0 | 0 | 0 | 0 | 0 | 0 | 0 | 0 | 0 | 0 | 0 | 0 | 0 | 0 | 0 |
| *Persistent slab* | 14 | -2 | 2 | -11 | -6 | 8 | 6 | 6 | 2 | 2 | 13 | 0 | -10 | -8 | -1 | -2 |
| *Persistent slab plus* | 1 | 1 | 0 | -1 | -1 | 0 | 0 | 0 | -1 | 1 | 0 | -1 | 0 | 0 | 1 | 1 |
| *Deep persistent slab* | 4 | -2 | 2 | -4 | -4 | -2 | -1 | -2 | -1 | 0 | 0 | 6 | 7 | 2 | 0 | 0 |
| *Spring-like* | 12 | -2 | -2 | 2 | 1 | -2 | -2 | 0 | -1 | 3 | -1 | -1 | 1 | 0 | 2 | 3 |
| *Loose wet & persistent slab* | 2 | 1 | -1 | -1 | 0 | 0 | 0 | 0 | 1 | 0 | 0 | -1 | 0 | 1 | -1 | -1 |

**Figure 10: Forecast region mean hazard situation prevalence and forecast region anomalies in percentage points for the alpine, treeline, and below treeline elevation band from 2011/12 to 2016/17. Red shading denotes positive anomalies, and blue shading indicates negative anomalies great than 5 percentage points.**



### 5.2.3 Seasonal snow climate classification

The results of the application of the Mock and Birkeland (2000) algorithm to the averages of the available weather observations over all seasons (2009/10 to 2016/17) generally agree with the traditional snow climate classification of the three main

mountain ranges (Fig. 11). Two of the three weather stations in the Coast Mountains were classified as maritime, while Blowdown Mid-Mountain, which is located in the Eastern section of the Coast Mountains (Duffy Lake region), was classified as transitional. Five of the six weather stations in the Columbia Mountains were assessed as having a transitional snow climate. The only non-transitional weather site in the Columbia Mountains was Kootenay Pass. This weather site was classified as maritime, which is consistent with its reputation as having larger amounts of new snow. All the weather stations in the Rocky

Mountains were classified as having a continental snow climate.

While the overall patterns confirm the existing snow climate classification, the winter-by-winter analysis revealed considerable variations in annual classifications. Within the study period, the 2014/15 and 2015/16 seasons emerged as the most maritime winters with more stations in the Columbia Mountains classified as maritime due to warmer average temperature and more rainfall. The 2016/17 season was the most continental winter with three weather stations in the Columbia Mountains, receiving

a continental classification due to strong December temperature gradients, and the two stations in the Coast Mountains being classified as transitional. The three winters 2011/12, 2012/13 and 2013/14 had overall a slightly more continental character with more continental classifications in the Coast and Columbia Mountains due to colder average air temperatures. During the study period, the 2009/10 and 2010/11 winters exhibited characteristics that were most similar to the overall snow-climate classification.



| | Elevation (relative location) | Overall | 2009/ 10 | 2010/ 11 | 2011/ 12 | 2012/ 13 | 2013/ 14 | 2014/ 15 | 2015/ 16 | 2016/ 17 |
|---|---|---|---|---|---|---|---|---|---|---|
| **Coastal Mountains** | | | | | | | | | | |
| Whistler | 1835 m (*mtn*) | Mari. (1) | Mari. (1) | Mari. (4) | n/a | n/a | Tran. (5) | Mari. (1) | Mari. (1) | Tran. (5) |
| Blowdown | 1890 m (*mid mtn*) | Tran. (5) | Tran. (7) | Tran. (5) | Tran. (5) | Tran. (7) | Tran. (5) | Mari. (1) | Tran. (5) | Tran. (5) |
| Little Bear | 1660 m (mtn) | Mari. (1) | Mari. (2) | Mari. (1) | Mari. (1) | Tran. (5) | Mari. (4) | Mari. (1) | Mari. (1) | Tran. (5) |
| **Columbia Mountains** | | | | | | | | | | |
| Sliding Mountain | 1675 m (mtn) | Tran. (7) | Tran. (7) | Tran. (7) | Tran. (7) | Tran. (7) | Tran. (7) | Mari. (1) | Mari. (2) | Cont. (3) |
| Sun Peaks | 2055 m (mtn) | Tran. (7) | n/a | n/a | Cont. (6) | Cont. (6) | Cont. (6) | Tran. (7) | Tran. (7) | Cont. (3) |
| Apex | 1750 m (mid mtn) | Tran. (7) | Tran. (7) | Tran. (7) | Tran. (7) | Tran. (7) | Cont. (3) | Mari. (2) | Tran. (7) | Cont. (3) |
| London Ridge | 2070 m (mtn) | Tran. (5) | Tran. (7) | Tran. (5) | Tran. (5) | n/a | Tran. (5) | Tran. (5) | Tran. (5) | Tran. (5) |
| Whitewater | 1950 m (mtn) | Tran. (7) | n/a | Tran. (5) | Tran. (5) | Tran. (5) | Tran. (5) | Tran. (7) | n/a | Tran. (5) |
| Kootenay Pass | 1780 m (mid mtn) | Mari. (1) | Tran. (5) | Mari. (1) | n/a | n/a | Tran. (5) | Mari. (1) | n/a | Mari. (1) |
| **Rocky Mountains** | | | | | | | | | | |
| Chatter Creek | 1615 m (valley) | Cont. (6) | Cont. (6) | n/a | Tran. (5) | Tran. (5) | Tran. (5) | n/a | n/a | Cont. (3) |
| Panorama | 2356 m (mtn) | Cont. (3) | Cont. (3) | Cont. (3) | Cont. (6) | Cont. (3) | Cont. (3) | Cont. (3) | Cont. (3) | Cont. (3) |
| Lake Louise | 2200 m (mtn) | Cont. (3) | Tran. (7) | Cont. (3) | n/a | Cont. (3) | Cont. (3) | Cont. (3) | Cont. (3) | Cont. (3) |
| Kananaskis | 1890 m (valley) | Cont. (3) | Cont. (3) | Cont. (3) | Cont. (3) | Cont. (3) | Cont. (3) | Cont. (3) | Cont. (3) | Cont. (3) |

**Figure 11: Overall and seasonal snow climate classifications according to Mock and Birkeland (2000): maritime (green), transitional (grey), continental (blue). The number in each field represents the snow climate decision. Seasons with insufficient weather observations are indicated with n/a.**

**6 Discussion**

**6.1 Typical hazard situations**

The identification of typical hazard situations represents an important step for quantitatively describing the nature of avalanche hazard conditions. While avalanche problems represent building blocks of avalanche hazard, the identified hazard situations can describe the complexity and severity of daily avalanche conditions much more comprehensive, but still concise way.

Our SOM analysis revealed twelve typical hazard situations that are combinations of the eight avalanche problem types identified in the CMAH (Statham et al., 2017). The twelve hazard situations and the additional *No avalanche problems* situation can roughly be grouped into four main classes: 1) situations typically associated with low danger ratings including the *No avalanche problems*, *Loose dry avalanche* and pure *Wind slab* hazard situations; 2) hazard situations dominated by storm slabs, which include pure *Storm slab* hazard situations and various combinations with wind slab and persistent slab avalanche

problems; 3) hazard situations with a dominant persistent avalanche problem (*Persistent slab*, *Persistent slab plus*, and *Deep persistent slab hazard situations*); and 4) hazard situations that occur during warmer conditions (*Spring-like* and *Loose wet & persistent slab hazard situation*). While the *No avalanche problems* situation was the most common hazard situation overall,





this situation rarely occurred in the alpine and treeline. The next most frequent situations were pure *Wind slab hazard situations*, pure *Storm slab hazard situations*, and *Persistent slab plus hazard situations*. Together these three hazard situations account for slightly more than one-third of the hazard situations across all seasons, forecast regions and elevation band.

## 6.2 Spatial variability in avalanche hazard conditions

We examined spatial variability of avalanche hazard in two dimensions, vertically by examining the differences between elevation bands and horizontally by examining the differences between forecast regions in Western Canada.

### 6.2.1 Elevation band differences in hazard conditions

The elevation band-specific prevalence values for the hazard situations exhibit expected patterns. All hazard situations including wind slab avalanche problems were considerably more prevalent in the alpine and at treeline. Similarly, the more
severe *Persistent slab plus* and *Deep persistent slab hazard situations* were more prevalent in the alpine and at treeline. However, pure *Storm slab*, the less severe *Persistent slab*, and the *Spring-like hazard situation* were considerably more prevalent below treeline. Together, these three hazard situations accounted for more than one-third of the hazard situations below treeline. The below treeline elevation band also had the highest frequency of *No avalanche problem* situations accounting for more than one third. Together, these results highlight that avalanche hazard conditions in the alpine and treeline
elevation bands are considerably more complex and severe than below treeline.

While conditions in the alpine and at treeline might differ on individual days, the prevalence of the different hazard situations across the entire study period was extremely similar between the two elevation bands. The biggest difference between these two elevation bands was that the prevalence of pure *Wind slab hazard situations*, which was 6 percentage points higher in the alpine than at treeline (20% versus 14%). The realism of these results nicely confirms the ability of the SOM approach to group
avalanche hazard situations into a set of meaningful patterns.

### 6.2.2 Regional differences in avalanche hazard conditions

Our comparison of the prevalence of typical hazard situations across the different forecast regions in Western Canada also revealed the expected patterns. Generally, the avalanche hazard conditions in forecast regions located in the *Coast Mountains* are dominated by pure *Wind slab* hazard situations and pure *Strom slab* hazard conditions. In the alpine elevation band, these
two hazard situations make up close 50% of the hazard conditions. Below treeline, *No avalanche problems* hazard situations comprise half of the assessments, and pure *Storm slab* hazard situations alone are responsible for approximately one quarter of the hazard situations in the South Coast region. On the other hand, the *Persistent slab plus* and *Deep persistent slab* hazard situation are much less frequent in these forecast areas. This picture generally agrees with the existing descriptions of the nature of avalanche hazard in the maritime snow climate of Coast Mountains (McClung and Schaerer, 2006). The hazard
situations are simpler (i.e., fewer simultaneous avalanche problems) and persistent avalanche problems are rare. If they occur, they are generally less severe than in the other climate zones.





In the *Columbia Mountains*, the snowpack gets more complex, and hazard situations that include persistent avalanche problems become more prevalent. Whereas the Cariboo and the North and South Columbia forecast regions exhibited higher prevalence value for *Storm & persistent slab* hazard situations, the Cariboo and the Purcell forecast regions also had more *Persistent slab plus* hazard situations. These observations are consistent with the perspective presented by Haegeli and McClung (2007) and

5 the general understanding of the transitional snow climate in Canada. The fact that the centrally located Glacier forecast region does not exhibit a similar increase in hazard situation involving persistent avalanche problems is a bit surprising. However, possible explanations for this deviation could be a) the unique geographic location of the forecast area, which is well known for its abundant snowfall (e.g., CCBFC (1995) cited in Haegeli and McClung, 2007), b) the fact that it is the only Parks Canada forecast region in the Columbia Mountains, or c) the relatively small size of the forecast region. Kootenay Boundary, the most

10 Southern forecast region in the Columbia Mountains also does not the higher prevalence of hazard situations involving persistent avalanche problems.

The most striking characteristic of the avalanche hazard conditions in the *Rocky Mountains* is the high prevalence of *Deep persistent slab* situations in the alpine and at treeline in the Central Rocky Mountain region. In the more southern forecast regions in the Rockies Mountains (South Rockies and Lizard Range), the dominance of the *Deep persistent slab* situations

15 disappears again and the *Persistent slab plus* (South Rockies) and pure *Storm slab* hazard situations (Lizard Range) become more prevalent. At treeline, the avalanche hazard characteristics of the Southern Rocky Mountains is similar to the Northwest Inland region in the Northern Coast Mountains. While this similarity might surprise at first, it does seem to make sense as these forecast regions exhibit avalanche hazard characteristics that are grounded in a continental snow climate but have strong maritime influences. This combination of continental and maritime influences is distinctly different from the traditional

20 transitional snow climate of the Columbia Mountains.

The observed hazard characteristics match the traditional perspective on the nature of avalanche hazard in the different mountain ranges in Western Canada quite well (e.g., McClung and Schaerer, 2006). At the same time, the hazard prevalence values indicate that there may be distinct sub-regions within the main mountain ranges, supporting the spatial variability of avalanche hazard described in previous studies (Gruber et al., 2004; Haegeli and McClung, 2007; Hägeli and McClung, 2003).

25 The most significant advancement of the approach presented in this study is, however, that it provides a much more detailed perspective on the type of avalanche hazard situations experienced in these regions and explicitly quantifies their prevalence. The quantitative nature of the characterization offers new opportunities for examining the observed differences statistically.

**6.3 Seasonal differences in avalanche hazard conditions**

Our comparison of the interseasonal variability in the snow climate classification of Mock and Birkeland (2000) and the

30 prevalence of the twelve hazard situations across Western Canada confirmed the results of Haegeli and McClung (2007) which showed that the nature of avalanche hazard can be dramatically different among winters that were classified similarly by the Mock and Birkeland (2000) algorithm. For example, the nature of avalanche hazard in the 2009/10 and 2010/11 winter varied dramatically even though the Mock and Birkeland (2000) algorithm assessed the two winters to be the most normal (i.e., the





most similar to the classification based on the average winter weather conditions during the entire study period). The 2009/10 winter was dominated by the *Persistent slab* hazard situations, whereas the 2010/11 had a higher prevalence of *Storm & wind slab* and *Storm, wind & persistent slab* hazard situations. Equally interesting is that the 2014/15 winter, which is one of the two most maritime winters, exhibited hazard situation prevalence values closest to the overall mean values for the entire study

period. However, there were also similarities between the snow climate scheme and the typical hazard prevalence values. For example, the winter 2015/16, the most maritime winter in the dataset, exhibited the highest seasonal prevalence of *Spring-like* hazard conditions.

These results highlight that examining the seasonal prevalence of typical hazard situations can offer a more insightful perspective on the avalanche hazard conditions of a winter than the Mock and Birkeland (2000) algorithm. Haegeli and

McClung (2007) already pointed out that the approach of Mock and Birkeland (2000) is limited because avalanches and their particular character are the result of specific sequences of weather events and not the average weather conditions of a winter. Whereas Haegeli and McClung (2007) simply used the number of persistent weak layers to characterize the nature of avalanche hazard of a winter, including all types of avalanche hazard situations into the analysis provides a much more complete and therefore meaningful perspective on what a winter was like.

**6.4 Limitations**

While the use of avalanche hazard assessments included in avalanche bulletins allowed us to characterize winters in a way that is more closely related to avalanche risk management than possible with weather observations, this dataset is not without challenges. Since the avalanche hazard assessments are human judgements, they are susceptible to human errors and biases as well as changes in operational procedures. For example, Avalanche Canada expressed that at the beginning of the 2012/13

winter, the forecaster team decided to no longer include Storm slab avalanche problems and Wind slab avalanche problems in the avalanche hazard assessments at the same time. This change in forecasting policy resulted in a general drop in the prevalence for Storm, wind, & persistent slab and Storm & wind hazard situations after the 2011/12 winter. Results including storm slab and wind slab avalanche problems should therefore be treated with caution. Geographic differences between forecast areas (e.g., size) as well as organizational and operational differences between Avalanche Canada and Parks Canada might

also cause systematic discrepancies among forecast regions unrelated to local weather and climate effects.

**7 Conclusion**

In this study, we present a new approach for describing the overall nature of avalanche hazard of a winter season. In contrast to previous approaches in this area, which used summarized seasonal weather observations, we used CMAH-based avalanche hazard assessments from Avalanche Canada and Parks Canada from the 2009/10 to 2016/17 winter seasons. Describing the

nature of an avalanche winter in a quantitative way required two distinct steps. First, we used SOM to identify typical avalanche hazard situations among the countless combinations of avalanche problems in the CMAH dataset. Second, we calculated the




overall prevalence of each typical hazard situation for the entire dataset as well as seasonal prevalence values for each forecast region and elevation band to describe the nature of the experienced avalanche hazard in a quantitative way.

Our research contributes to the existing literature on avalanche climate and its variability is multiple ways. First, the identification of typical hazard situations and the calculation prevalence values provide an innovative approach for describing

the nature of seasonal avalanche hazard conditions. The resulting measure is concise and conducive to statistical analyses, but still provides a comprehensive picture that is informative for avalanche risk management due to its link to avalanche problem types. Summarizing the nature of avalanche hazard this way has the potential to open new opportunities for studying the effects of large scale climate oscillations and climate change on avalanche hazard. Most of the existing studies on the effect of atmospheric oscillations on avalanche hazard (Dixon et al., 1999; Keylock, 2003; McClung, 2013; Reardon et al., 2008;

Thumlert et al., 2014) and climate change on avalanche hazard (Bellaire et al., 2016; Castebrunet et al., 2012; Jamieson et al., 2017; Laternser and Schneebeli, 2002; Lazar and Williams, 2008; Sinickas et al., 2016) have focused on examining trends in historical avalanche activity records. While avalanche activity along transportation corridors is tightly monitored, variations in avalanche control practices can make it difficult to attribute observed changes to long-term changes in winter weather conditions (Bellaire et al., 2016; Jamieson et al., 2017; Sinickas et al., 2016). Avalanche datasets of backcountry operators are

inherently incomplete as areas are large and reduced visibility often prevents visual inspection (Hägeli and McClung, 2003; Laternser and Schneebeli, 2003). Dendrochronological methods date the occurrence of extreme avalanche events, but it is difficult obtaining large enough datasets and extreme events can only provide a limited perspective on the character of the overall avalanche activity (Hebertson and Jenkins, 2003). The advantage of basing the description of the seasonal nature of avalanche hazard on expert assessments is that it represents a comprehensive measure of operational concerns that ingrates a

wide range of relevant observations. This partially circumvents the issue of incomplete and noisy dataset that are common in avalanche safety research. At the same time, avalanche assessment practices can change over time, which makes long-term studies challenging. However, efforts to model avalanche problem characteristics from observed and modeled weather and snowpack observations might offer a new pathway for examining the effect of climate change on avalanche hazard in an informative way.

The results of our study also provide insight into the large-scale spatial and temporal variability of avalanche hazard in Western Canada with an unprecedented level of detail. Overall, the avalanche hazard patterns that emerged from our analysis generally align with expected patterns. Similar to Haegeli and McClung (2007), our methodology reveals that the nature of seasonal avalanche hazard can vary dramatically among winters that are classified the same by the Mock and Birkeland (2000) snow climate classification algorithm. For elevation band hazard patterns, the alpine and treeline elevation bands have a similar

hazard pattern with greater prevalence of wind slab avalanche problems, more severe persistent slab avalanche problems and deep persistent slab avalanche problems. The below treeline elevation band exhibits less complex hazard situations including considerably more *No avalanche problems* situations. Our comparison across the different forecast regions in Western Canada revealed the expected patterns with pure *Wind slab* or *Storm slab* hazard situations comprising half of the hazard conditions in the traditional maritime snow climate zone. The avalanche hazard becomes more complex in the Columbia Mountains with



hazard situations with persistent slab avalanche problems becoming more prevalent. The high prevalence of *Deep persistent slab* situations in the Central Rocky Mountain forecast region is the most prominent regional hazard variance of this study. Our study also showed that the Southern Rocky Mountain regions and Northwest Inland region exhibit similar avalanche situation characteristics. While geographically distant, the similarity can be attributed to the fact that both forecast regions are

5  grounded in a continental snow climate with strong occasional maritime influences.

Future research in this area would benefit from also including hazard assessments from U.S. public avalanche warning services, which have also broadly adopted the CMAH. Expanding the geographic extent of the analysis would provide broader insight into the spatial variability of avalanche hazard in North America. Hazard assessments according to the CMAH from datasets like the InfoEx of the Canadian Avalanche Association might offer new opportunities for looking at smaller scale variabilities

10  in the nature of avalanche hazard. However, differences in the interpretation and application of the CMAH among operations might severely limit these possibilities.



**9 Acknowledgements**

We would like to recognize the valuable insight of James Floyer and Karl Klassen from Avalanche Canada and Grant Statham from Parks Canada, and also for providing the archived public bulletins for this study. The NSERC Industrial Research Chair in Avalanche Risk Management at Simon Fraser University (SFU) is financially supported by Canadian Pacific Railways, Helicat Canada, the Canadian Avalanche Association and Mike

Wiegele Helicopter Skiing. The SFU Avalanche Research Program is further supported by Avalanche Canada and Avalanche Canada Foundation. Bret Shandro was also supported by a Mitacs Accelerate fellowship in partnership with Avalanche Canada.

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
