# Peer review of "Title page"

_Natural Hazards and Earth System Sciences, 2018_

## Referee Comment (RC1) · C. Mock (Referee) · 4 Feb 2018

Overall this is a very good paper, I suggest publication with some minor revisions. The authors have a unique avalanche dataset in W Canada to study their objectives on snow avalanche characteristic very closely. Methods are robust and results are very well described, including the SOM (Self-organizing maps), and figures in the most part are very good. Figures and tables a bit numerous, more so than the average manuscript, they are useful and present a lot of information, but I wonder how perhaps some can be condensed. It is hard for me to comment further at this point. Maybe more information can be added to Supplemental material online. Below are some comments.

1) The authors are correct on the top of p. 3 noting the little research on specifying the

[Figure]

nature of avalanches (loose wet slabs, etc.) as well as avalanche safety aspects within the context of broad climate types. The authors should note that Mock and Birkeland (2000) and some of their subsequent papers did note the role of short-term weather conditions within days to a few weeks (particularly as driven by synoptic patterns) on faceted crystals etc., so Mock and Birkeland did not simply just rely on the 3 main climate specifications. Thus assertions on interseasonal variability are not too surprising (bottom p. 26). The authors on p. 27, should strike out the limitation of Mock and Birkeland on ignoring the sequences of weather events, as they acknowledged the importance of studying shorter weather timescales.

2) The authors should note Fitzharris' older Rogers Pass work. Though old, it's the most comprehensive long-term avalanche record in W Canada dating back to the 19th century and has been studied extensively in its climate context.

3) p. 12, Table 1. On the SOM hazard situations (which is a very good research approach), I assume that avalanche control is not a factor in creating any of these?, and that the data set is truly unique with regards to avalanche control (which the authors describe later in the period that is usually a problem with avalanche data sets).

4) Section 5.2.2. Again I assume spatial variability of hazard situations are not impacted by aspects of avalanche control? This includes the high prevalence of deep persistence slabs in the alpine and treeline of the Central Rocky Mountain region.

5) Biggest thing missing in the analyses, and the logical next step would be to have a synoptic meteorological/climate component to relating with the appropriate hazard situations (ex. storm & wind slab events, spatial variability of hazard situations etc.), and if available, some snow pit data analyses. This should stress timeframes shorter than ENSO/seasonal, given the avalanche hazard situation findings. The large-scale meteorological analyses and forecast model simulations in some part, with recognition on non-weather factors etc. may provide additional forecast value to help in terms of probability, and perhaps explain some additional differences in seasons as presented

in Figures 6-8 (Seasonal prevalence of typical hazard situations). This can be stressed a bit strongly briefly in a few places in the paper.

6) Conclusion: the authors on p. 28 correctly point out the issues and difficulty in long term avalanche datasets and practices, and applications in the backcountry. They seem a bit too negative on long-term studies overall, and this should just be brief mention and be more constructive/complimentary, as long-term approaches are often studied differently with those issues in mind. That includes dendrochronological studies, which have very good value on probability of events, etc. Here, citing the Fitzharris studies show the value and perhaps an example of looking at things at a longer time-frame, and merging different interdisciplinary approaches.
* * *

---

## Author Comment (AC1) · 13 Feb 2018

**Overall this is a very good paper, I suggest publication with some minor revisions. The authors have a unique avalanche dataset in W Canada to study their objectives on snow avalanche characteristic very closely. Methods are robust and results are very well described, including the SOM (Self-organizing maps), and figures in the most part are very good.**

We would like to thank Dr. Mock for his positive assessment of our manuscript and his constructive suggestions.

**Figures and tables a bit numerous, more so than the average manuscript, they are useful and present a lot of information, but I wonder how perhaps some can**

[Figure]

**be condensed. It is hard for me to comment further at this point. Maybe more information can be added to Supplemental material online.**

We agree with the reviewer that this study includes a lot of figures, but we but we find it difficult to present the richness of the data and the results in a more concise way. However, to address the reviewer's concerns and to reduce the figure and table overload in the main manuscript, we decided to move the SOM error figure and the prevalence anomaly figures (Fig. 7, 8 &10) into supplemental material. To provide readers with a reference interpreting the variability within the prevalence values directly out of the bar charts, we added and additional bar representing mean prevalence values on the very left of Fig. 6 & 9.

**Below are some comments.**

1. **The authors are correct on the top of p. 3 noting the little research on specifying the nature of avalanches (loose wet slabs, etc.) as well as avalanche safety aspects within the context of broad climate types. The authors should note that Mock and Birkeland (2000) and some of their subsequent papers did note the role of short-term weather conditions within days to a few weeks (particularly as driven by synoptic patterns) on faceted crystals etc., so Mock and Birkeland did not simply just rely on the 3 main climate specifications. Thus assertions on interseasonal variability are not too surprising (bottom p. 26). The authors on p. 27, should strike out the limitation of Mock and Birkeland on ignoring the sequences of weather events, as they acknowledged the importance of studying shorter weather timescales.**

We acknowledge that the Mock and Birkeland (2000) include December snowpack temperature gradient to account for faceting of the entire early-season snowpack, a process common in continental snow and avalanche climates. In our opinion, this still

represents an average condition since the average December temperature gradient applies to the entire snowpack and the resulting weak snowpack foundation typically persists for the entire season. This seems to be distinctly different from the more short-term fluctuations in weather conditions that are the primary drivers for direct action avalanches and the formation of persistent weaknesses, such as surface hoar layers.

We did not intend to say that Mock and Birkeland (2000) did not acknowledge that the effect of short-term weather fluctuations on avalanche activity, but rather that the existing avalanche and snow climate definitions do not take them into account. To clarify our intent and more accurately represent the contribution and perspective of Mock and Birkeland (2000), we made the following changes to our manuscript:

- P. 3 - Lines 10-17: Described the limitation of the snow climate classification more explicitly and included a brief description of Mock and Birkeland's (2000) additional trend analysis.

The newly developed algorithm allowed Mock and Birkeland (2000) to examine spatial and interseasonal variabilities in snow and avalanche climate characteristics in the western United States and explore the potential effects of El Nino Southern Oscillation, the **Pacific-North American Pattern** and the Pacific Decadal Oscillation on avalanche conditions. **While their seasonal snow climate classifications showed whether winters associated with different phases of these climate oscillations were more continental or more maritime as a whole, it did not provide more detailed insight about the nature of avalanche hazard during those winters. However, their temporal trend analysis of seasonal avalanche indices at select sites with reliable long-term avalanche activity records in the central Rocky Mountains showed some interesting relationship between the magnitude of avalanche activity and climate oscillations**.

- P. 3 - Line 23: Added Mock and Birkeland (2000) as an additional reference high-

lighting the importance of short-term weather fluctuations on avalanche activity.

. . . seasonally summarized weather observations only have limited connections to the factors driving daily avalanche hazard. Instead, avalanche hazard is determined by short-term weather fluctuations and particular sequences of weather events that dominate over general climate effects (Gruber et al., 2004; **Mock and Birkeland, 2000** ).

- P. 25 - Lines 8-15: We slightly reworded the section to better highlight that we refer to the snow climate classification algorithm and not to the entire study by Mock and Birkeland (2000).

These results highlight that examining the seasonal prevalence of typical hazard situations can offer a more insightful perspective on the avalanche hazard conditions of a winter than the **snow climate classification algorithm of** Mock and Birkeland (2000). **While the classification schema considers early season faceting, a common situation in continental snow climates that affects the nature of avalanche hazard for the entire rest of a season**, it is limited because avalanches and their particular character are the result of specific sequences of weather events and not the average weather conditions of a winter.

2. **The authors should note Fitzharris' older Rogers Pass work. Though old, it's the most comprehensive long-term avalanche record in W Canada dating back to the 19$^{th}$ century and has been studied extensively in its climate context.**

We agree with Dr. Mock, that work of Fitzharris (1987) should be mentioned in this study and we apologize for the oversight. We have added citations to this study in several locations in the manuscript including:

- On page 2 line 13 & 32

- On page 28 line 10.

3. **p. 12, Table 1. On the SOM hazard situations (which is a very good research approach), I assume that avalanche control is not a factor in creating any of these?, and that the data set is truly unique with regards to avalanche control (which the authors describe later in the period that is usually a problem with avalanche data sets).**

4. **Section 5.2.2. Again I assume spatial variability of hazard situations are not impacted by aspects of avalanche control? This includes the high prevalence of deep persistence slabs in the alpine and treeline of the Central Rocky Mountain region.**

This assumption is correct. The uniqueness of our dataset is that avalanche problem characterizations included in the bulletins a) represent assessments for the uncontrolled backcountry (i.e., not affected by changes in avalanche control practices), and b) are spatially a larger-scale assessment than the more point-location perspective that avalanche observations of transportation corridors offer. To better highlight these features of our dataset, we added additional explanations in several locations in the manuscript including:

- On page 4 line 23

The resulting dataset of avalanche problem assessments for **uncontrolled backcountry areas** across large areas of western Canada offers new opportunities . . . .

- On page 4 line 23

The core information presented in Canadian avalanche bulletins consists of a characterization of avalanche problems **in uncontrolled backcountry areas** according to the CMAH . . .

- On page 6 line 8

The combined dataset from Avalanche Canada and Parks Canada consists of . . . forecast regions that **comprehensively cover the main mountain ranges in western Canada** (Tables S1 & S2).

5. **Biggest thing missing in the analyses, and the logical next step would be to have a synoptic meteorological/climate component to relating with the appropriate hazard situations (ex. storm & wind slab events, spatial variability of hazard situations etc.), and if available, some snow pit data analyses. This should stress timeframes shorter than ENSO/seasonal, given the avalanche hazard situation findings. The large-scale meteorological analyses and forecast model simulations in some part, with recognition on nonweather factors etc. may provide additional forecast value to help in terms of probability, and perhaps explain some additional differences in seasons as presented in Figures 6-8 (Seasonal prevalence of typical hazard situations). This can be stressed a bit strongly briefly in a few places in the paper.**

We completely agree with Dr. Mock that the logical next step is to explore climate affects on avalanche hazard by correlating the hazard variability mentioned in this study with climate variability. While our study actually examined some of these relationships and identified some interesting correlations, we decided that including all results in a single manuscript would be too overwhelming. Hence, the present manuscript focuses

on the method for identifying hazard patterns, while a second manuscript (currently in preparation and soon to be submitted to NHESS) will focus on the relationship with climate oscillations.

6. **Conclusion: the authors on p. 28 correctly point out the issues and difficulty in long term avalanche datasets and practices, and applications in the backcountry. They seem a bit too negative on long-term studies overall, and this should just be brief mention and be more constructive/complimentary, as long-term approaches are often studied differently with those issues in mind. That includes dendrochronological studies, which have very good value on probability of events, etc. Here, citing the Fitzharris studies show the value and perhaps an example of looking at things at a longer timeframe, and merging different interdisciplinary approaches.**

We completely agree Dr. Mock that studies using long-term avalanche records (e.g., Dixon et al., 1999; Fitzharris, 1987; Hebertson and Jenkins, 2003; Keylock, 2003; Mc-Clung, 2013; Reardon et al., 2008; Thumlert et al., 2014; Bellaire et al., 2016; Castebrunet et al., 2012; Jamieson et al., 2017; Laternser and Schneebeli, 2002; Lazar and Williams, 2008; Sinickas et al., 2016) have provided many useful results and have inspired creative research examining the problem of characterising avalanche hazard variability. We believe that the various approaches offer perspectives with different strengths and weaknesses and that all perspectives are necessary to develop a comprehensive understanding. To clarify our position, we have reworded parts of the conclusion on page 26.

---

## Referee Comment (RC2) · Dr. Sokratov (Referee) · 14 Feb 2018

The paper presents interesting approach of combination of "avalanche climates" and the snow avalanche hazard characteristics based on extensive database of the catalogued actual snow avalanches in western Canada. Results of similar "comprehensive" studies, going a bit further, were reported in past (i.e. Miagkov S.M. Kanaev L.A. (Eds.) Geografiia lavin [Geography of avalanches], Moscow: Izdatel'stvo Moskovskogo universiteta, 1992. 331p.). Also, there are several classifications of the snow avalanches produced in Russia, including the "genetic classification" of V.V.Dzuba presented in the book cited above, where the types of avalanches were related to meteorology and stratigraphy of snow cover, definitely related to the conclusions in the presented manuscript. But it was published in Russian, used different climatic parameters and

was mainly focused on the territory of USSR. Any comparison would probably have purely academic value. The paper provides detailed review of the gradual development of the snow avalanche hazard assessment, not touching the risk evaluation side. In my view this should be expressed in the abstract and text more clearly. Risk is mentioned several times. Or, at least ideas on how different "avalanche problems" would affect the value of risk should be presented. There is such link in Statham et al. (2018) (the year in citations should be corrected), but that paper is only cited as a source of CMAH. Strange not to see citation and links to "A seasonal snow cover classification system for local to global applications" by Sturm et al. in the "avalanche climate types" discussion. Evidently different "classes of snow covers" should affect the avalanche hazard. Very interesting is the analysis of the seasonal (is not it inter-annual?) variability in the prevalence of various snow avalanches hazard situations (Figure 6). Not quite clear what the anomalies in percent means in Figures (Tables?) 7 and 8 for the comparison of different years though. Same applies to Figure (Table?) 10. Despite these notes the paper is really good and the presented approach can indeed be used for other regions, where such an extensive dataset on snow avalanches is available. And in my view this is the main limitation of the approach - suitable only for well-documented sites. My suggestion is to publish the paper as it is or with minor editions.

---

## Referee Comment (RC3) · Anonymous Referee #3 · 16 Feb 2018

This is a very interesting paper with a unique database of avalanche hazards for different geographic location of western Canada. I really appreciate the innovative methodology (self-organizing maps) and the robustness of the results and related figures and tables. In that regards, most of them present useful information although some might be considered as supplemental material.

Considering the high quality of the submitted paper, I only have a few general comments. As mentioned, the inter-annual variability is not surprising and clearly shows the importance to have a deeper look at the synoptic situations leading to an increased avalanche hazard. It also demonstrates the limitations of the snow avalanche climate classification and related avalanche hazard for risk management. In this regard, and considering the importance of storms for avalanche problems, it could be interesting in

the next future to look at the ratio from different storm tracks and 500-mbar composite anomaly maps such as reported by Martin and Germain (2017).

I also completely agree with the authors concerning the need of good quality and specific data to improve our knowledge. However, I suggest adding one or two sentences in the discussion section about the availability of the weather data and the extrapolation based on a few weather stations. Do you think a more robust network of weather stations could significantly improve the delimitation of avalanche climate? Also, in the Conclusion section, the authors stated the need for looking at smaller scale variabilities but also to include the U.S. hazard assessments. However, because snow avalanches are mainly driven by climate at various spatiotemporal scales, it should also be stated the need for better climate variability analysis such as teleconnexion (PDO, El Nino and so on) but also in order to detect trends, if any, in climate variability and what might explain this intra- and inter-annual variability.

Please also note the supplement to this comment:
https://www.nat-hazards-earth-syst-sci-discuss.net/nhess-2018-2/nhess-2018-2-RC3-supplement.pdf

---

## Author Comment (AC2) · 17 Feb 2018

The paper presents interesting approach of combination of "avalanche climates" and the snow avalanche hazard characteristics based on extensive database of the catalogued actual snow avalanches in western Canada. Results of similar "comprehensive" studies, going a bit further, were reported in past (i.e. Miagkov S.M. Kanaev L.A. (Eds.) Geografiia lavin [Geography of avalanches], Moscow: Iz-datel'stvo Moskovskogo universiteta, 1992. 331p.). Also, there are several classifications of the snow avalanches produced in Russia, including the "genetic classification" of V.V.Dzuba presented in the book cited above, where the types of avalanches were related to meteorology and stratigraphy of snow cover, definitely related to the conclusions in the presented manuscript. But it was pub-

**lished in Russian, used different climatic parameters and was mainly focused
on the territory of USSR. Any comparison would probably have purely academic
value.**

We would like to thank Dr. Sokratov for the revealing perspective into the field of
avalanche climate studies in Russia, however it is unfortunate that this work is not
available in English.

**The paper provides detailed review of the gradual development of the snow
avalanche hazard assessment, not touching the risk evaluation side. In my view
this should be expressed in the abstract and text more clearly. Risk is mentioned
several times. Or, at least ideas on how different "avalanche problems" would af-
fect the value of risk should be presented. There is such link in Statham et al.
(2018) (the year in citations should be corrected), but that paper is only cited as
a source of CMAH.**

While the physical risk from avalanches is managed by continuously monitoring haz-
ard conditions and choosing mitigation measures accordingly to reduce the risk to el-
ements of value exposed to the existing hazard (e.g., infrastructure, buildings, people)
to an acceptable level. Hence, risk only comes into play when the hazard interacts
with elements of value and is therefore context dependent. Since our study examines
the nature of avalanche hazard in Western Canada separate from the existing ele-
ments of value, risk is not included in our discussion. However, as pointed out by Dr.
Sokratov, we reference publications (e.g., Statham et al., 2018) that describe the link
of avalanche hazard to risk in detail. We believe that this is appropriate.

We would like to thank Dr. Sokratov for pointing out the error in our citations of Statham
et al. (2018). We fixed the issue throughout the manuscript.

**Strange not to see citation and links to "A seasonal snow cover classification
system for local to global applications" by Sturm et al. in the "avalanche climate
types" discussion. Evidently different "classes of snow covers" should affect**

**the avalanche hazard.**

We thank Dr. Sokratov for pointing out this shortcoming. Sturm et al. (1995) was an informative study for the Mock and Birkeland (2000) snow climate classification scheme. To address this issue, we added some text and a Sturm et al. (1995) citation at the beginning on page 3 on line 3:

> Building on previous research, Mock and Birkeland (2000) introduced a classification algorithm that objectively classifies the local snow and avalanche climate of individual winter seasons ... The authors derived classification thresholds by analyzing the variabilities of the select input parameters (Sturm et al., 1995) at locations with established snow-climate classifications (Armstrong and Armstrong, 1986).

**Very interesting is the analysis of the seasonal (is not it inter-annual?) variability in the prevalence of various snow avalanches hazard situations (Figure 6). Not quite clear what the anomalies in percent means in Figures (Tables?) 7 and 8 for the comparison of different years though. Same applies to Figure (Table?) 10.**

Dr. Sokratov is correct to point out our word choice regarding winter to winter or interseasonal variability of typical avalanche prevalence values was imprecise. We therefore changed the caption of Fig. 6 (page 18) to, "Interseasonal prevalence of typical hazard situations." and adjusted the heading of section 5.2.1 to "Typical hazard situation prevalence: Interseasonal variability"

In response to the comments from the first reviewer (Dr. Mock) and reduce redundancy, we strengthened the prevalence bar charts (Fig. 6 and 9) by adding an additional bar representing mean prevalence values on the very left and we moved the prevalence anomaly figures (Figures 7, 8 & 10) into supplemental material. The intent of these anomaly figures it to explicitly highlight the interseasonal variability in prevalence values that are hard to visually extract directly from the bar charts. To clarify our intent, we

have now added an additional sentence in the text before referring to the first anomaly figure (page 17, line 7):

> . . . The seasonal anomaly values represent the percent difference between the seasonal prevalence value and the mean prevalence value for each of the thirteen avalanche hazard situations during the study period (Fig. S2). . . .

To highlight positive and negative anomalies of temporal and spatial variability of hazard situation prevalence values we added red and blue shading to indicate positive and negative anomalies great than 5 percentage points respectively. Because of the coloring, we assumed that NHESS would be laying out these tables as figures. We are happy to oblige with whatever is most appropriate for NHESS.

**Despite these notes the paper is really good and the presented approach can indeed be used for other regions, where such an extensive dataset on snow avalanches is available. And in my view this is the main limitation of the approach - suitable only for well-documented sites. My suggestion is to publish the paper as it is or with minor editions.**

We would like to thank Dr. Sokratov for his positive assessment of our manuscript and his insightful suggestions.

---

## Referee Comment (RC4) · Dr. Sokratov (Referee) · 26 Feb 2018

In my view "it is unfortunate that this work is not available in English" should be changed on "it is unfortunate we were not able to assess the data not available in English", but this is ok. I support publishing the paper with the changes accepted.

---

## Author Comment (AC3) · 26 Feb 2018

**This is a very interesting paper with a unique database of avalanche hazards for different geographic location of western Canada. I really appreciate the innovative methodology (self-organizing maps) and the robustness of the results and related figures and tables. In that regards, most of them present useful information although some might be considered as supplemental material.**

We would like to thank this anonymous reviewer for the positive assessment of our manuscript and their suggestions towards future research questions. Your comment regarding moving material into the supplementary section aligns with previous comments by Dr. Mock and Dr. Sokratov. We decided to move the SOM error figure (Fig.

[Figure]

4) and the prevalence anomaly figures (Fig. 7, 8 &10) into the supplemental material. To enable readers to extract information on the seasonal and regional variabilities of the hazard situation prevalence values directly out of the bar charts (Fig. 6 & 9), we added an additional bar representing mean prevalence values on the very left as a reference.

**Considering the high quality of the submitted paper, I only have a few general comments. As mentioned, the inter-annual variability is not surprising and clearly shows the importance to have a deeper look at the synoptic situations leading to an increased avalanche hazard. It also demonstrates the limitations of the snow avalanche climate classification and related avalanche hazard for risk management. In this regard, and considering the importance of storms for avalanche problems, it could be interesting in the next future to look at the ratio from different storm tracks and 500-mbar composite anomaly maps such as reported by Martin and Germain (2017).**

We support the reviewer's assertion that investigating storm track variability is a useful approach for exploring the effects of seasonal climate variability on avalanche hazard. We believe that this would be an interesting next step for investigation.

**I also completely agree with the authors concerning the need of good quality and specific data to improve our knowledge. However, I suggest adding one or two sentences in the discussion section about the availability of the weather data and the extrapolation based on a few weather stations. Do you think a more robust network of weather stations could significantly improve the delimitation of avalanche climate?**

While additional weather stations allow for more local summaries following the Mock & Birkeland (2000) classification scheme, we believe this would not significantly improve the characterization of avalanche hazard because this requires a meaningful link between average weather observations and the nature of avalanche hazard. In the

introduction and discussion, we cover the limitations of using weather station observations for characterizing avalanche hazard.

On page 3 line 23:

*This result highlighted that the existing snow climate classifications have considerable limitations for informing avalanche risk management practices. This is not surprising as seasonally summarized weather observations only have limited connections to the factors driving daily avalanche hazard. Instead, avalanche hazard is determined by short-term weather fluctuations and particular sequences of weather events that dominate over general climate effects (Gruber et al., 2004; Mock and Birkeland, 2000).*

On page 25 line 8:

These results highlight that examining the interseasonal prevalence of typical hazard situations can offer a more insightful perspective on the avalanche hazard conditions of a winter than the snow climate classification algorithm of Mock and Birkeland (2000). While the classification schema considers early season faceting, a common situation in continental snow climates that affects the nature of avalanche hazard for the entire rest of a season, it is limited because avalanches and their particular character are the result of specific sequences of weather events and not the average weather conditions of a winter.

**Also, in the Conclusion section, the authors stated the need for looking at smaller scale variabilities but also to include the U.S. hazard assessments. However, because snow avalanches are mainly driven by climate at various spatiotemporal scales, it should also be stated the need for better climate variability**

**analysis such as teleconnexion (PDO, El Nino and so on) but also in order to detect trends, if any, in climate variability and what might explain this intra- and inter-annual variability.**

We completely agree with the anonymous reviewer that the logical next step is to explore the effects of climate variabilities on avalanche hazard by correlating the inter-annual hazard variability mentioned in this study with climate oscillation indices (ENSO, PDO, PDA, AO). Our study actually includes the analysis of some of these relationships and identified some interesting correlations but including all our results in a single manuscript would have been too overwhelming. Hence, the present manuscript focuses on the method for identifying hazard patterns, while a second manuscript (currently in preparation and soon to be submitted to NHESS) will focus on the relationship with climate oscillations.

---

## Author Comment (AC4) · 7 Mar 2018

While the work of Miagkov & Kanaev (1992) is available, we are unable to include this work as it is not available in English.